# A Strategy to Provide a Present and Future Scenario of Mexican Biodiversity of Tardigrada

Jazmín García-Román [1,2], Alba Dueñas-Cedillo [1,2], Montserrat Cervantes-Espinoza [1,2], José Juan Flores-Martínez [3], Carlos Fabián Vargas-Mendoza [4], Enrico Alejandro Ruiz [2,*] and Francisco Armendáriz-Toledano [1,*]

1 Colección Nacional de Insectos, Instituto de Biología, Universidad Nacional Autónoma de México, Cto. Zona Deportiva S/N, C.U., Ciudad de Mexico C.P. 04510, Mexico; lgarciar0706@alumno.ipn.mx (J.G.-R.); albaduenas@live.com.mx (A.D.-C.); montbio20@gmail.com (M.C.-E.)

2 Laboratorio de Ecología, Departamento de Zoología, Escuela Nacional de Ciencias Biológicas, Instituto Politécnico Nacional, Prolongación de Carpio y Plan de Ayala S/N, Ciudad de Mexico C.P. 11340, Mexico

3 Laboratorio de Sistemas de Información Geográfica, Instituto de Biología, Universidad Nacional Autónoma de México, Cto. Zona Deportiva S/N, C.U., Ciudad de Mexico C.P. 04510, Mexico; jj@ibunam.mx

4 Laboratorio de Variación Biológica y Evolución, Departamento de Zoología, Escuela Nacional de Ciencias Biológicas, Instituto Politécnico Nacional, Prolongación de Carpio y Plan de Ayala S/N, Ciudad de Mexico C.P. 11340, Mexico; carfvargas@yahoo.com

* Correspondence: eruizc@ipn.mx (E.A.R.); farmendariztoledano@ib.unam.mx (F.A.-T.)

**Abstract:** Although the number of known tardigrade taxa in Mexico has increased significantly in the last ten years, the knowledge of their diversity faces challenges, as more than half of the Mexican territory has no records of this phylum. Thus, we developed a strategy to provide a present and future scenario for understanding the Mexican biodiversity of Tardigrada, described the distribution patterns of the current recorded species, calculated the estimated richness, and the estimated taxonomic effort needed to complete the national inventory. We obtained 474 records of 105 taxa, belonging to 42 genera and 75 species, distributed in 12 of the 14 biogeographical provinces of Mexico. We found that 54.72% of the species are present in more than three world regions and 3.79% of species that have been recorded only in Mexican provinces. Distribution patterns could be recognized for 11 species, two of which have a Nearctic distribution, seven are Neotropical and two are distributed in both regions. The Mexican biogeographical provinces with the greatest diversity of tardigrades, both at specific and generic level, were the Transmexican Volcanic Belt (TVBP) and the Sierras Madre Oriental (SMOrP) and Sierra Madre Occidental (SMOcP), which have been previously identified as particularly species-rich regions. Diversity estimation methods predict that more than 290 species of tardigrades could be found in Mexico.

**Keywords:** Mexican tardigrades; richness estimation; Clench model; accumulation curve

## 1. Introduction

Tardigrades are micrometazoan between 50 and 1200 μm in length, ubiquitous in marine, freshwater, and terrestrial interstitial communities, and have been found in diverse habitats [1–3]. They are among the most desiccation- and radiation-tolerant animals surviving even to extreme levels of ionizing radiation [4–7]. Given the great variety of environments in which they can live, tardigrades have been required to allocate significant amounts of energy toward specific adaptive strategies to survive, such as resting stages, generally defined as dormant, stages involving a temporary suspension of active life, reduced or suspended metabolism, and arrested development [8]. The great resilience of these organisms has allowed them to disperse through air and water, and recently, possible evidence shows that birds could play a role as long-distance dispersers of tardigrades have been found [9].

The study of the diversity of this phylum has been addressed through inventories [10–13], ecological studies at different geographic scales [14–17], and integrative

taxonomic studies [18–20]. As a result of these efforts, at least 150 genera and more than 1300 species are now recognized worldwide [21]. Of these species, most were described exclusively with morphological attributes and only 293 species have sequences recorded in GeneBank (CO I, CO II, ITS1, ITS2, 16S 18S, and 28S) [22]; therefore, the use of integrative analyses could allow the recognition of cryptic species in the future.

Mexico ranks fourth in terms of species richness, with 10% of the world's species living within its territory, which corresponds to just over one percent of the earth's surface [23]. Based on the listing of various biological groups, in 2014, around 94,000 species were documented in Mexico, and it was estimated that due to the number of yet undescribed species, this amount could increase more than three-fold for vertebrates and surprisingly from four to more than 20-fold for invertebrates [24]. Although these estimations include numerous taxa, they did not consider at least ten animal phyla, including Tardigrades, among them [25]. Until 2011, the knowledge of tardigrades in Mexico was based on very restricted and occasional sampling, in some cases not even intended to address this taxon, with collections that did not include habitat descriptions and/or distribution patterns at local or regional scales [26–32]. However, studies over the last 10 years include species listings [33–36], new records [37–44], and descriptions of new species [45–49]. The results of these researchers suggest that, by the year 2021, in Mexico, more than 70 tardigrade species were recognized, and most of them are limnoterrestrial moss dwellers [43] and to a lesser extent marine species. Almost half of the species recognized in the country were documented in less than three years, from 44 species in 2018 [25], up to more than 70 in 2021 [43], which stands for an important increase in the records of the group.

One of the main reasons for studying diversity is the increasing trend of species loss, which has led to the recognition that we are experiencing the sixth mass extinction [24]. Species richness estimation studies play a fundamental role in the in-depth knowledge of diversity [50]. Therefore, to achieve a good representation of species in a region, a great sampling effort is required, especially in very diverse assemblages with low abundant taxa [51], as is the case of tardigrades.

The knowledge of the number of species in a region and the description of their distribution patterns are imperative for an understanding of the relationships between species and the environment, detecting spatial patterns at different scales, identifying biogeographic patterns, supporting biogeographic regionalization [52], delimiting areas of high diversity [53], and for the creation of optimal conservation strategies [54–57].

Although the number of tardigrade taxa in Mexico has increased significantly in the last ten years, the knowledge of their diversity faces challenges, as more than half of the Mexican territory has no records of the phylum [25,47]. Moreover, there are very few taxonomists dedicated to the study of this group, and the documented species lacks corresponding molecular data which, along with descriptions and an analysis of their distribution patterns, could allow for comparisons of global diversity and to determine whether species are endemic or widely distributed. In this study, we developed a strategy to provide a present and future scenario of Mexican biodiversity of Tardigrada. We list the Mexican tardigrades, describe species distribution patterns of the species under different geographic scenarios, calculate the estimated richness of tardigrades in the country, and estimated the taxonomic effort needed to complete the national inventory.

## 2. Materials and Methods

### 2.1. List of the Mexican Tardigrades

First, we determined how many species have been described and recorded in Mexico. An exhaustive literature search was conducted including not only indexed journals but also conference proceedings, theses, and scientific outreach journals (popular media). We built a database with information on locality, geographic coordinates, species, and year of publication of the literature. We use the Google Earth platform to assign georeferenced to records without geographic coordinates.

## *2.2. Distribution Patterns of the Tardigrades of Mexico*

The biogeographic patterns exhibited by the taxa present in Mexico show a large complexity. In this regard, various hypotheses have been raised to explain the assembly of different taxa with diverse origins [58]. Thus, to determine the biogeographic provinces in which the tardigrades are distributed, all records at the genus and species level were projected on the layer of Mexican biogeographic provinces by Morrone et al. [59], and the marine ecoregions [60]. To evaluate in a global context the distribution patterns of the taxa found, worldwide information was integrated using Morrone's biogeographic regionalization [61], which is organized into kingdoms, regions, subregions, and domains. For Mexico, the regionalization into Mexican provinces was used [59]. With this information, it was determined whether the species were found in small regions or are broadly distributed (i.e., present in more than three biogeographic regions). We also described the number of records and species per biogeographic province and for each taxon, the provinces where their records were found and identified.

## *2.3. Estimating Richness*

There are plenty of methods to estimate species richness, making it difficult to conduct direct comparisons among them. Therefore, we decided to follow the work of Stork [62], who provided a framework and a reference of existing methods. In his work, seven estimation methods are mentioned; however, given the nature of our available dataset (a diverse range of data from literature; with no field collections was made) only two of them were applied: (1) ratio of known to unknown species, and (2) accumulation curves.

### 2.3.1. Ratio of Known to Unknown Species

This method is used to estimate the number of species of a poorly known group ("problem taxon") in a determinate geographic region, based on the proportion of species of other well-documented taxa in the same region, concerning their diversity worldwide. To do this, we used the information on the number of species of widely studied groups. A proportion or percentage of known species for those taxa in Mexico was obtained concerning the rest of the world; with the percentages obtained for each taxon, an extrapolation was made of the total number of tardigrade species recognized worldwide [62]. In this way, well studied, representative groups that have a number of species present in Mexico and a number of species worldwide were selected; in this case, we started from a Mexican and global known diversity of 15 eukaryote groups, which in turn were obtained from articles and databases: vascular plants [63,64], mosses [65,66], birds [67,68], reptiles [69,70], mammals [71,72], fishes [73,74], arthropods [75,76], proturans [77,78], diplurans [78,79], copepods [80,81], onychophorans [82,83], nematodes [84], rotifers [85] and cladocerans [86]. The number of currently described species of Tardigrada was obtained from the Actual Checklist of Tardigrade 40th ed [21]. Likewise, a global estimate of tardigrades was made by Bartels et al. [50]; these data were also taken as a second reference of estimated tardigrade diversity. In this way, two references were obtained, the first for current diversity with a value of 1380 species and the second one for estimated diversity with a value of 2654 species [50]. Hence, the use of these values were applied as follows: if Rotifera has 3000 known species worldwide, and 184 rotifer species are reported in Mexico, then 6.1% of the global rotifer diversity is present in Mexico. If this ratio is extrapolated to phylum Tardigrada, based on the 1380 described species, 6.1% of this diversity represents 84 species in Mexico, but if we use the richness estimation by Bartels et al. [50] (n = 2654), we could expect as much as 162 species. The calculation for the rest of the phyla was performed in this way. Since most of the 1380 species of tardigrades known to date have been described from morphological data, and current integrative approaches uphold that these attributes are not sufficient to support and recognize them, it is likely that there are synonymous or cryptic species in this number. For this reason, calculations obtained from methods that consider the current number of species should be taken as a conservative estimate of the possible Mexican tardigrade diversity.

2.3.2. Accumulation Curves

This method is based on the estimation of the predicted asymptote for the cumulative number of species over time. It uses the information of the presence and frequency of new species over time, so the closer the curve is to the asymptote, the more difficult it is to find new taxa. From the literature previously consulted in the list of Mexican tardigrades (in the first section of the methods), collection data were extracted from the sites where tardigrades were recorded, since the accumulation curve method uses standardized sampling units and in the reviewed works this unit is not defined, and so each geographical locality corresponding to a collection event was taken as a sampling unit, recognizing a total of 136 units from 25 bibliographic sources. Given that in some studies the taxonomic assignation was carried out only at a genus level, two accumulation curves were elaborated (genus and species categories). For the curve at the genus level, the data from the collections identified to species and genus were combined. However, it should also be noted that this curve was obtained using data from sources other than indexed journals, such as theses, scientific outreach, conference proceedings, and posters, which in turn tend to be perceived as taxonomically "less rigorous" [35,37,39,41,87,88]. Taxonomic modifications of the taxa, as synonymies or genus changes, were also updated from the species checklist [21].

The curves were estimated following the protocol of Jiménez-Valverde and Hortal [89], which represents the increase of the taxa in the inventory according to the performed sampling. This generates an "ideal curve", which is based on randomization of the sampling effort and the mean number of species [90]. Subsequently, the quality of the sampling $n = a/(1 + b \cdot n)^2$ and the sampling effort required to record 95% of the fauna $n_{95} = 0.95/[b (1 - 0.95)]$ were calculated with the *a* and *b* constants of Clench's equation $S_n = a \cdot n/(1 + b \cdot n)$. This model assumes that the probability of adding new species to the inventory increases as more time is spent in the field, and is recommended for both large study areas and protocols [91].

*2.4. Taxonomic Effort*

Taxonomic effort is a function of search rate (collection effort) and species description (taxonomic description effort) [92]. The first one was calculated from accumulation curves (see above). Taxonomic description effort implies the amount of effort, after collection, invested in describing new species; this task is carried out by specialized taxonomists [93]. Therefore, in this section, the taxonomic identification effort was calculated as the number of specialists that would be needed to describe potential species. To estimate the taxonomic effort needed to complete the inventory of the Mexican tardigrades, the taxonomic effort made in the rest of the world was evaluated as a reference that could be applied to the Mexican scenario. To this end, we evaluated the relationship between the number of species described and the number of authors participating in this taxonomic activity worldwide since 1834. All the species, their year of description, and corresponding authorities were listed from the 40th edition of the actual checklist of tardigrade species [21]; from this list, we obtained the average time spent in the taxonomic field (how many years an author remains describing species) and the average number of species described by each author.

On the other hand, based on the results of the species estimation methods, the number of species missing to complete the inventory was calculated and the rate of species description in the world and Mexico was estimated. We do not attempt to assign a deterministic value to the number of authors and the time required to inventory the Mexican tardigrade fauna, since these variables depend on numerous factors that were not considered in the present study (e.g., the level of experience of taxonomists, the number of research groups and their infrastructure—national or international—, the economic resources invested, the rate of habitat loss), those are factors that could considerably affect the rate of species description. Rather, it should be considered that our estimates of potential diversity are based on biased samples from a few regions within the country and that most of the Mexican provinces have few or no records, particularly provinces characterized by their high

species richness, which in turn could contribute significantly to the discovery of new taxa of Tardigrades in the country.

## 3. Results

### 3.1. List of the Mexican Tardigrades

We quantified 474 tardigrade records (Table 1). One of them was identified only at Order (Eutardigrada), one at subfamily (Florarctinae), one at the family level (Halechiniscidae), 238 at the genus level, and 233 at the species level. These records correspond to 105 taxa from 18 families, 7 subfamilies, 42 genera, and 75 species. Seven genera were recorded more than 30 times: *Macrobiotus* (70), *Minibiotus* (53), *Hypsibius* (53), *Adropion* (43), *Diphascon* (42), *Milnesium* (40), and *Ramazzottius* (38); seven genera had from ten up 24 records: *Echiniscus* (23), *Paradiphascon* (20), *Doryphoribius* (15), *Paramacrobiotus* (14), *Famelobiotus* (10), *Mesocrista* (10), and *Milnesioides* (10); and 24 taxa were recorded from one up eight times: *Mesobiotus* (8), *Pilatobius* (6), *Pseudechiniscus* (6), *Coronarctus* (5), *Dipodarctus* (4), *Archechiniscus* (3), *Cornechiniscus* (3), *Florarctus* (3), *Isohypsibius* (3), *Styraconyx* (3), *Astatumen* (2), *Batillipes* (2), *Viridiscus* (2), *Wingstrandarctus* (2), *Anisonyches* (1), *Calcarobiotus* (1), *Calohypsibius* (1), *Dactylobiotus* (1), *Diaforobiotus* (1), *Dianea* (1), *Echiniscoides* (1), Eutardigrada (1), Florarctinae (1), *Guidettion* (1), *Halechiniscus* (1), Halechiniscidae (1), *Haplomacrobiotus* (1), *Itaquascon* (1), *Kristenseniscus* (1), *Megastygarctides* (1), and *Paratanarctus* (1).

**Table 1.** List of Tardigrada taxa recorded in Mexico up to this review. Terra typica, worldwide records and distribution, records in Mexico and Mexican biogeographic provinces. Transmexican Volcanic Belt Province (TVBP), Sierra Madre Oriental Province (SMOrP), Yucatán Peninsula Province (YP), Sierra Madre Occidental Province (SMOcP), Tamaulipas Province (TP), Veracruzan Province (VP), Pacific Lowlands Province (PLP), Chihuahuan Desert Province (ChDP), Baja California Province (BCP), Balsas Basin Province (BBP), Sierra Madre del Sur Province (SMSP), Chiapas Highlands Province (ChHP), Northern Gulf of Mexico (NGM) and Caribbean Sea (CS).

| Taxa | Terra Typica | Worldwide Records and Distribution | Records in Mexico | Mexican Biogeographic Provinces |
|---|---|---|---|---|
| **Heterotardigrada Doyère, 1840** | | | | |
| **Order Arthrotardigrada Marcus, 1927** | | | | |
| **Family Anisonychidae Møbjerg, Jørgensen & Kristensen, 2019** | | | | |
| *Anisonyches* sp. Pollock, 1975 | - | Persic Gulf, Indic Ocean, Atlantic, Caribbean [94,95]. **Widely distributed** | Yucatán [38] | YP (1) |
| **Family Archechiniscidae Binda, 1978** | | | | |
| *Archechiniscus bahamensis* Bartels, Fontoura & Nelson, 2018 | Bahamas | Caribbean [94,95]. **Neotropical** | Caribbean Sea [46] | CS (2) |
| *Archechiniscus* sp. Schulz, 1953 | - | Atlantic, Caribbean, Coral Sea, Indic Ocean, Ionian Sea, Pacific Ocean, Tasman Sea [94,95]. **Widely distributed** | Yucatán [37] | YP (1) |
| **Family Batillipedidae Ramazzotti, 1962** | | | | |
| *Batillipes* sp. Richters, 1909 | - | Europe, Asia [94], North America [95]. **Holarctic** | Yucatán [37,38] | YP (2) |
| **Family Coronarctidae Renaud-Mornant, 1974** | | | | |
| *Coronarctus mexicus* Romano III, Gallo, D'Addabbo, Accogli, Baguley & Montagna, 2011 | Gulf of Mexico | North America [96]. **Only registered near of the type locality** | Gulf of Mexico [38] | NGM (1) |

| Taxa | Terra Typica | Worldwide Records and Distribution | Records in Mexico | Mexican Biogeographic Provinces |
|---|---|---|---|---|
| *Coronarctus* sp. Renaud-Mornant, 1974 | - | Pacific Ocean, Atlantic Ocean, Gulf of México [94,95]. **Widely distributed** | Gulf of Mexico [38] | NGM (1) |
| **Family Halechiniscidae Thulin, 1928** | | | | |
| *Dipodarctus* cf. *subterraneus* (Renaud-Debyser, 1959) | Atlantic Ocean | Pacific Ocean, Atlantic Ocean, Mediterranean Sea, Caribbean. [94,95] **Widely distributed** | Caribbean Sea, Yucatán [46] | CS (1), YP (1) |
| *Dipodarctus* sp. Pollock, 1995 | - | Pacific Ocean, Caribbean, Atlantic Ocean, Mediterranean Sea [94,95]. **Widely distributed** | Caribbean Sea, Yucatán [37,46] | CS (1), YP (1) |
| Florarctinae sp. Renaud-Mornant, 1982 | - | Pacific Ocean, Atlantic Ocean, Caribbean [94,95]. **Holarctic** | Caribbean Sea [46] | CS (1) |
| *Florarctus yucatanensis* Anguas-Escalante, Navarrete, Demilio, Pérez-Pech & Hansen, 2020 | Mexico | North America [48]. **Only registered in the type locality** | The Caribbean Sea, Quintana Roo [48] | CS (1), YP (2) |
| Halechiniscidae sp. Thulin, 1928 | - | Pacific Ocean, Atlantic Ocecan, Indic Ocean [94,95]. **Widely distributed** | Yucatán [46] | YP (1) |
| *Halechiniscus* cf. *perfectus* Schulz, 1955 | Mediterranean Sea | Mediterranean Sea, Caribbean Sea [94]. **Only registered in the type locality** | Quintana Roo [37] | Unknown |
| *Wingstrandarctus corallinus* Kristensen, 1984 | Australia | North America, Australia, Caribbean [94,95]. **Widely distributed** | Caribbean Sea [33] | CS (1) |
| *Wingstrandarctus* sp. Kristensen, 1984 | - | Pacific Ocean, Atlantic Ocean, Caribbean [94,95]. **Widely distributed** | Yucatán [37] | YP (1) |
| **Family Stygarctidae Schulz, 1951** | | | | |
| *Megastygarctides* sp. McKirdy, Schmidt & McGinty-Bayly, 1976 * | - | India, Caribbean [94,95]. **Tropical** | Caribbean Sea [44] | CS (1) |
| **Family Styraconyxidae Kristensen & Renaud-Mornant, 1983** | | | | |
| *Paratanarctus* sp. D'Addabbo Gallo, Grimaldi de Zio, Morone De Lucia & Troccoli, 1992 * | - | The Mediterranean Sea, Caribbean [94,95]. **Palearctic-Neotropical** | Caribbean Sea [46] | CS (1) |
| *Styraconyx robertoi* Pérez-Pech, Navarrete, Demilio, Anguas-Escalante & Hansen, 2020 | Mexico | North America [46]. **Only registered in the type locality** | Caribbean Sea, Yucatán [46] | CS, YP |
| **Order Echiniscoidea Richters, 1926** **Family Echiniscidae Thulin, 1928** | | | | |
| *Cornechiniscus lobatus* (Ramazzotti, 1943) | Italy | North America [97], South America [98], Africa [99]. **Widely distributed** | Tamaulipas, Coahuila [25,32,86] | TP (1), SMOrP (2) |
| *Echiniscoides* sp. Plate, 1888 * | - | North America [97], Africa [99], Europe, Asia, Oceania 100]. **Widely distributed** | Yucatán [37] | YP (1) |
| *Echiniscus becki* Schuster & Grigarick, 1966 | USA | North America [97]. **Only registered in the type locality** | Baja California [41] | BCP (1) |

**Table 1.** *Cont.*

| Taxa | Terra Typica | Worldwide Records and Distribution | Records in Mexico | Mexican Biogeographic Provinces |
|---|---|---|---|---|
| *Echiniscus blumi* Richters, 1903 | Spitsbergen | North America [97], Europe [100], Africa [99]. **Widely distributed** | Baja California [41] | BCP (1) |
| *Echiniscus* cf. *tamus* Mehlen, 1969 | USA | North America [97]. **Nearctic** | Chihuahua, Nuevo León [25,28] | SMOrP (1), SMOcP (1) |
| *Echiniscus kerguelensis* Richters, 1904 | Kerguelen Is. | Europe, Antarctic, India [100], North America [97], Africa [99]. **Widely distributed** | Mexico State, Morelos [29,86] | TVBP (2) |
| *Echiniscus manuelae* da Cunha & du Nascimento Ribeiro, 1962 | Portugal | Europe [100], North America [97], Central America [101], South America [98]. **Widely distributed** | Nuevo León [25] | SMOrP (1) |
| *Echiniscus siergristi* Heinis, 1911 | Mexico | North America [97]. **Only registered in the type locality** | Oaxaca [26,30] | LPP (1) |
| *Echiniscus* sp. C.A.S. Schultze, 1840 | - | Europe, India, Antarctic, Oceania [100], North America [97], Africa [99]. **Widely distributed** | Oaxaca, Hidalgo, Mexico City, Quintana Roo [26,35,36,42,44,45] | SMOrP (2), TVBP (10), YP (2) |
| *Kristenseniscus kofordi* (Schuster & Grigarick, 1966) | Santa Cruz Is. (Ecuador) | North America [97], Central America [101], South America [98]. **American** | Chiapas [31] | VP (1) |
| *Pseudechiniscus* cf. *juanitae* de Barros, 1939 | Brazil | North America [97], Central America [101], South America [98]. **American** | Chiapas, Nuevo León [25,31] | TP, (1), LPP (1), VP (1) |
| *Pseudechiniscus facettalis* Petersen 1951 | Switzerland | Europe, Asia, Australia [100], North America [97], Central America [101], South America [98], Africa [99]. **Widely distributed** | Chihuahua [28] | SMOcP (1) |
| *Pseudechiniscus quadrilobatus* Iharos, 1969 | Vietnam | North America [97]. **Only registered in the type locality** | Chiapas [31] | VP (1) |
| *Pseudechiniscus* sp. Thulin, 1911 | - | North America [97], South America [98], Africa [99], Europa, Asia [100]. **Widely distributed** | Chihuahua [28] | SMOcP (1) |
| *Pseudechiniscus suillus* (Ehrenberg, 1853) | Switzerland | North America [97], Europa [100], Africa [99]. **Widely distributed** | Oaxaca [26] | SMSP (1) |
| *Viridiscus viridis* (Murray, 1910) | Oahu Is. (Hawaii) | North America [97] South America [98], Europe [100]. **Widely distributed** | Chihuahua [28] | SMOcP (1) |
| *Viridiscus viridissimus* (Péterfi, 1956) | Romania | Europe [100], North America [97], South America [98]. **Widely distributed** | Oaxaca [33] | BBP (1) |
| **Class Eutardigrada Richters, 1926** | | | | |
| Eutardigrada sp. Richters, 1926 | - | Europe, Asia, Oceania [100], North America [97], Central America [99], South America [98], Africa [99]. **Widely distributed** | Hidalgo [45] | TVBP (1) |
| **Order Apochela Schuster, Nelson, Grigarick & Christenberry, 1980** | | | | |
| **Family Milnesiidae Ramazzotti, 1962** | | | | |
| *Milnesioides* sp. Claxton, 1999 * | - | Australia [101], North America [97]. **Australian-Nearctic** | Mexico City [35] | TVBP (10) |

**Table 1.** *Cont.*

| Taxa | Terra Typica | Worldwide Records and Distribution | Records in Mexico | Mexican Biogeographic Provinces |
|---|---|---|---|---|
| *Milnesium barbadosense* Meyer & Hinton, 2012 * | Barbados | North America [97], South America [98]. **American** | Nuevo León, Tamaulipas [25,49] | SMOrP (7), TP (1) |
| *Milnesium cassandrae* Moreno-Talamantes, Roszkowska, García-Aranda, Flores-Maldonado & Kaczmarek, 2019 | Mexico | North America [97]. **Only registered in the type locality** | Nuevo León, Coahuila, Tamaulipas [25,49] | TP (9) |
| *Milnesium fridae* Moreno-Talamantes, León-Espinosa, García-Aranda, Flores-Maldonado & Kaczmarek, 2020 | Mexico | North America [97]. **Only registered in the type locality** | Nuevo León [49] | TP (2) |
| *Milnesium* sp. Doyère, 1840 | - | Europe, Asia, Oceania [100], North America [97], Central America [101], South America [98], Africa [99]. **Widely distributed** | Baja, California, Nuevo León, San Luis Potosí, Mexico City, Hidalgo, Quintana Roo [34,35,41–44,49,87] | BCP (1), ChDP (2), SMOrP (1), TVBP (1), YP (1) |
| *Milnesium tardigradum* Doyère, 1840 | France | Europe, Asia, Oceania [100], North America [97], Central America [101], South America [98], Africa [99]. **Widely distributed** | Chihuahua, Morelos, Chiapas, Mexico State [28,29,33] | SMOcP (6), TVBP (2), VP (1) |
| **Order Parachela Schuster, Nelson, Grigarick & Christenberry, 1980** | | | | |
| **Family Calohypsibiidae Pilato, 1969** | | | | |
| *Calohypsibius* cf. *ornatus* (Richters, 1900) | Ireland | North America [97], South America [98]. **Widely distributed** | Mexico State [35] | TVBP (1) |
| **Family Doryphoribiidae Gąsiorek, Stec, Morek & Michalczyk, 2019** | | | | |
| *Doryphoribius chetumalensis* Pérez-Pech, Anguas-Escalante, Cutz-Pool & Guidetti, 2017 | Mexico | North America [97]. **Only registered in the type locality** | Quintana Roo [45] | YP (4) |
| *Doryphoribius dawkinsi* Michalczyk & Kaczmarek, 2010 * | Costa Rica | North America [97], Central America. [101] **American** | Nuevo León [25] | SMOrP (2) |
| *Doryphoribius evelinae* (Marcus, 1928) | Germany | Europe [100], North America [97], South America [98]. **Widely distributed** | Chihuahua [28,30] | SMOcP (1) |
| *Doryphoribius flavus* (Iharos, 1966) | Hungary | Europe [100], North America [97], Central America [101]. **Widely distributed** | Chiapas [31] | VP (1) |
| *Doryphoribius gibber* Beasley & Pilato, 1987 | USA | North America [97], Asia [100]. **Holarctic** | Chiapas [31] | VP (1) |
| *Doryphoribius mexicanus* Beasley, Kaczmarek & Michalczyk, 2008 | Mexico | North America [97]. **Only registered in the type locality** | Oaxaca [29] | SMSP (1) |
| *Doryphoribius quadrituberculatus* Kaczmarek & Michalczyk, 2004 * | Costa Rica | North America [97], Central America. [101] **American** | Nuevo León [25] | TP (1) |

**Table 1.** *Cont.*

| Taxa | Terra Typica | Worldwide Records and Distribution | Records in Mexico | Mexican Biogeographic Provinces |
|---|---|---|---|---|
| *Doryphoribius* sp. Pilato, 1969 | - | North America [97], South America [98], Europe, Asia [100], Africa [99]. **Widely distributed** | Quintana Roo [36,45] | YP (5) |
| *Paradiphascon* sp. Dastych, 1992 * | - | Africa [99], North America [97]. **Tropical-Nearctic** | Mexico City [35] | TVBP (20) |
| **Family Halobiotidae Gąsiorek, Stec, Morek & Michalczyk, 2019** | | | | |
| *Dianea sattleri* (Richters, 1902) | Germany | Europe, Asia, Oceania [100], North America [97], Central America, [101] South America [98], Africa [99]. **Widely distributed** | Chiapas [31] | VP (1) |
| *Haplomacrobiotus hermosillensis* May 1948 | Mexico | North America [97]. **Only registered in the type locality** | Sonora [27,30] | SMOcP (1) |
| **Family Hypsibiidae Pilato, 1969** | | | | |
| *Adropion scoticum* (Murray, 1905) | Sweden | Europe [100], North America [97], South America [98], Africa [99]. **Widely distributed** | Mexico State [47] | TVBP (3) |
| *Adropion* sp. Pilato, 1987 | - | North America [97], South America [98], Africa [99], Europe, Asia [100]. **Widely distributed** | Mexico City [35] | TVBP (40) |
| *Astatumen trinacriae* Arcidiacono, 1962 | Italy | Europe [100], North America [97], South America [98], Africa [99]. **Widely distributed** | Nuevo León [25] | SMOrP (1), DChP (1) |
| *Diphascon chilenense* Plate, 1888 | Chile | North America [97], South America [98]. **American** | Chihuahua [28] | SMOcP (1) |
| *Diphascon mitrense* Piato, Binda, & Qualtieri, 1999 | Argentina | North America [97], South America [98]. **American** | Mexico State [47] | TVBP (2) |
| *Diphascon pingue* (Marcus, 1936) | Germany | North America [97], Central America [101], South America [98], Africa [99], Europe, Asia, Antarctic [100]. **Widely distributed** | Nuevo León, Coahuila, Mexico State [25,47] | TVBP (4), SMOrP (3), ChDP (1) |
| *Diphascon* sp. Plate, 1888 | - | North America [97], South America [98], Antarctica, Europe, Asia [100], Africa [99]. **Widely distributed** | Hidalgo, Mexico City [35,44] | TVBP (31) |
| *Guidettion carolae* (Binda & Pilato, 1969) | Italy | Europe [100] and North America [95]. **Holarctic** | Nuevo León [25] | SMOrP (1) |
| *Hypsibius* cf. *convergens* (Urbanowicz, 1925) | Lithuania | Europe, Asia, Antarctic, Oceania [100], Africa [99], North America [97], South America [98]. **Widely distributed** | Chihuahua [28] | ChDP (1), SMOcP (4) |
| *Hypsibius* cf. *microps* Thulin, 1928 | Sweden | Europe, Asia [100], North America [97], South America [98], Africa [99]. **Widely distributed** | Mexico State [47] | TVBP (4) |
| *Hypsibius* cf. *pallidus* Thulin, 1911 | Sweden | Europe, Asia [100], North America [97], South America [98], Africa [99]. **Widely distributed** | Mexico State [30,47] | TVBP (3) |

**Table 1.** *Cont.*

| Taxa | Terra Typica | Worldwide Records and Distribution | Records in Mexico | Mexican Biogeographic Provinces |
|---|---|---|---|---|
| *Hypsibius* sp. Ehrenberg, 1848 | - | North America [97], South America [98], Antarctica, Europe [100], Africa [98]. **Widely distributed** | Mexico City, Mexico State [35] | TVBP (40) |
| *Isohypsibius sculptus* (Ramazzotti, 1962) | Chile | Europe [100], South America [98], North America [97]. **Widely distributed** | Morelos [29] | TVBP (1) |
| *Isohypsibius* sp. Thulin, 1928 | - | North America [97], Europa, Asia, Antarctica [100], Africa [99]. **Widely distributed** | Hidalgo [44] | SMOrP (1) |
| (?) *Isohypsibius* sp. Thulin, 1928 | - | North America [97], Europa, Asia, Antarctica [100], Africa [99]. **Widely distributed** | Chihuahua [28] | SMOcP (1) |
| *Itaquascon umbellinae* de Barros, 1939 | Brazil | North America [97], South America [98], Africa [99]. **Widely distributed** | Chihuahua [28] | SMOcP (1) |
| *Pilatobius nodulosus* Ramazzotti, 1957 | USA | North America [97]. **Nearctic** | Mexico State [29,47] | SMOrP (2), TVBP (3) |
| **Family Macrobiotidae Thulin, 1928** | | | | |
| *Calcarobiotus* cf. *polygonatus* (Binda & Guglielmino, 1991) | Tanzania | Africa [99], North America [97]. **Tropical-Nearctic** | Nuevo León [39] | SMOrP (1) |
| *Famelobiotus* sp. Pilato, Binda & Lisi, 2004 * | - | Asia [100], North America [97]. **Oriental-Nearctica** | Mexico City [35] | TVBP (10) |
| *Macrobiotus alvaroi* Pilato & Kaczmarek, 2007 | Costa Rica | Central America [101], North America [97]. **American** | Chiapas [33] | VP (1) |
| *Macrobiotus anemone* Meyer, Domingue & Hinton, 2014 * | USA | North America [97]. **Nearctic** | Tamaulipas [25] | TP (1) |
| *Macrobiotus ascensionis* Richters, 1908 | Ascension Is. | Atlantic Ocean [94,95]. **Only registered in the type locality** | Unknown [30] | |
| *Macrobiotus* cf. *acadianus* (Meyer & Domingue, 2011) * | USA | North America [97]. **Nearctic** | Nuevo León [25] | TP (1) |
| *Macrobiotus echinogenitus* Richters, 1903 | Spitsbergen (Norway) | Europe, Asia [100], Africa [99], North America [97], South America. [98] **Widely distributed** | Mexico State [29] | TVBP (1) |
| *Macrobiotus hufelandi* C.A.S. Schultze, 1834 | Germany | Europe, Asia [100], North America [97], Central America [101], South America [98], Africa [99]. **Widely distributed** | Mexico State, Oaxaca, Chihuahua [26,28,29,41] | TVBP (1), LPP (1), SMOcP (1) |
| *Macrobiotus kazmierskii* Kaczmarek & Michalczyk, 2009 * | Argentina | North America [97] South America. [98] **American** | Nuevo León [25] | TP (1) |
| *Macrobiotus ocotensis* Pilato, 2006 | Mexico | North America [97]. **Only registered in the type locality** | Chiapas [102] | VP (1) |
| *Macrobiotus persimilis* Binda & Pilato 1972 * | Italy | Europe [100], North America [97]. **Holarctic** | Chiapas [33] | ChHP (1) |
| *Macrobiotus rubens* Murray, 190 | Himalayas (India) | India, Australia [100], North America [97], South America [98], Africa [99]. **Widely distributed** | Oaxaca [26,30] | LPP (1) |

**Table 1.** *Cont.*

| Taxa | Terra Typica | Worldwide Records and Distribution | Records in Mexico | Mexican Biogeographic Provinces |
|---|---|---|---|---|
| *Macrobiotus* sp. C.A.S. Schultze, 1834 | - | Europe, India, Antarctic, Oceania [100], North America [97], Africa [99]. **Widely distributed** | Oaxaca, Hidalgo, Mexico City, Mexico State, Quintana Roo [26,30,42,44,47] | SMOrP (4), TVBP (45), LPP (1) YP (2) |
| *Macrobiotus terminalis* (Bertolani & Rebecchi, 1993) | Italy | Europe [100] North America [97]. **Holarctic** | Oaxaca [33,36] | BBP (1) |
| *Mesobiotus contii* Pilato & Lisi, 2006 | Mexico | North America [97]. **Only registered in the type locality** | Chiapas [31] | VP |
| *Mesobiotus coronatus* (de Barros 1942) | Brazil | South America [98], Central America [101], North America [97], Europa [100], Africa [99]. **Widely distributed** | Oaxaca, Chihuahua [28,33] | BBP (1), SMOcP (2) |
| *Mesobiotus diffusus* (Binda & Pilato, 1987) * | Tunisia (Africa) | Africa [99], North America [97]. **Tropical-Nearctic** | Mexico State, Coahuila [39,88] | TVBP (1), SMOrP (1) |
| *Mesobiotus harmsworthii* (Murray, 1907) | Franz Joseph Land (Russia) | Europe [100], North America [97], Central America [101], South America [98], Africa [99]. **Widely distributed** | Sinaloa, Oaxaca [26,29] | LPP (1), SMOcP (2) |
| *Mesobiotus* sp. Vecchi, Cesari, Bertolani, Jönsson, Rebecchi & Guidetti, 2016a | - | North America [97], South America [98], Europe, Asia [100], Africa [99]. **Widely distributed** | Hidalgo, Mexico City, Quintana Roo [34,42,44,87] | SMOrP (1), TVBP (2), YP (1) |
| *Mesocrista* sp. Pilato, 1987 | - | Europe, Asia [100], North America [97]. **Holarctic** | Mexico City [35] | TVBP (10) |
| *Minibiotus* cf. *intermedius* (Plate, 1888) | Chile | Europe, Asia [100], South America [98], North America [97]. **Widely distributed** | Chihuahua, Nuevo León [28] | SMOcP (1) |
| *Minibiotus citlalium* Dueñas-Cedillo & García-Román, 2020 in Dueñas-Cedillo et al. 2020 | Mexico | North America [97]. **Only registered in the type locality** | Mexico State [47] | TVBP (1) |
| *Minibiotus continuus* Pilato & Lisi, 2006 | Mexico | North America [97]. **Only registered in the type locality** | Chiapas [31] | TP (1), VP (1), SMOrP (2) |
| *Minibiotus furcatus* (Ehrenberg, 1859) * | Switzerland | Europe [100], North America [97]. **Holarctic** | Mexico State, Morelos [29] | TVBP (1) |
| *Minibiotus sidereus* Pilato, Binda & Lisi, 2003 | Ecuador | South America [98], North America [97]. **American** | Mexico State [47] | TVBP (6) |
| *Minibiotus* sp. R.O. Schuster, 1980 | - | North America [97], South America [98], Antarctica, Europe, Asia, Oceania [100], Africa [99]. **Widely distributed** | Mexico City, Nuevo León [35,39] | TVBP (40), SMOrP (1) |
| *Paramacrobiotus* cf. *klymenki* Pilato, Kiosya, Lisi & Sabella, 2012 * | Ukraine | Europe [100], North America [97]. **Holarctic** | Nuevo León [39] | SMOrP (1) |
| *Paramacrobiotus areolatus* (Murray, 1907) | Svalbard (Norway) | Europe [100], North America [97], Central America [101], South America [98], Africa [99]. **Widely distributed** | Chihuahua [28,45] | SMOcP (1) |

**Table 1.** *Cont.*

| Taxa | Terra Typica | Worldwide Records and Distribution | Records in Mexico | Mexican Biogeographic Provinces |
|---|---|---|---|---|
| *Paramacrobiotus richtersi* (Murray, 1911) | Ireland | Europe [100], North America [97], Central America [101], South America [98], Africa [99]. **Widely distributed** | Nuevo León [28] | SMOcP |
| *Paramacrobiotus* sp. Guidetti, Schill, Bertolani, Dandekar & Wolf, 2009 | - | North America [97], South America [98], Europe [100], Africa [99]. **Widely distributed** | Hidalgo, Mexico, City, Nuevo León [35,41,87] | TVBP (10), SMOrP (3) |
| **Family Microhypsibiidae Pilato, 1998** | | | | |
| *Ramazzottius baumanni* (Ramazzotti, 1962) | Chile | North America [97], Central America [101], South America [98], Australia [100]. **Widely distributed** | Mexico State, Michoacán, Morelos [29] | TVBP (3) |
| *Ramazzottius* cf. *oberhaeuseri* (Doyère, 1840) | Germany | Europe, Asia, Oceania [100], North America [97], South America [98], Africa [99]. **Widely distributed** | Mexico State, Michoacán, Nuevo León [25,29,34] | TP (1), SMOrP (1), TVBP (2) |
| *Ramazzottius* sp. Binda & Pilato, 1986 | - | North America [97], Antarctica, Europe [100], Africa [99]. **Widely distributed** | Baja California, Mexico City, Quintana Roo [35,43] | BCP (1), TVBP (30), YP (1) |
| **Family Murrayidae Guidetti, Rebecchi & Bertolani, 2000** | | | | |
| *Dactylobiotus parthenogeneticus* Bertolani, 1982 * | Italy | Europe [100], North America [97], South America [98]. **Widely distributed** | Nuevo León [39] | SMOrP (1) |
| **Family Richtersiusidae Guidetti, Schill, Giovannini, Massa, Goldoni, Ebel, Förschler, Rebecchi & Cesari, 2021** | | | | |
| *Diaforobiotus islandicus* (Richters, 1904) | Iceland | North America [97], Central America [101], Europa, Asia [100], Africa [99]. **Widely distributed** | Nuevo León [40] | SMOrP |

* The known distribution range of these taxa was expanded. (?) the author is not sure of the genus [28].

According to the bibliographic sources, we obtained 311 of the records from theses, 151 from indexed journals, nine from conference proceedings, and three from scientific outreach journals [35,37,39,41,87,88].

### 3.2. Distribution Patterns of the Tardigrades of Mexico
Geographic Distribution

The distribution map of records (Figure 1) showed that tardigrades have been collected in 12 of the 14 biogeographic provinces: 317 records from the Transmexican Volcanic Belt Province (TVBP), 42 in the Sierra Madre Oriental Province (SMOrP), 26 in the Yucatán Peninsula Province (YP), 26 in the Sierra Madre Occidental Province (SMOcP), 20 in the Tamaulipas Province (TP), 11 in the Veracruzan Province (VP), 5 in the Pacific Lowlands Province (PLP), 5 in the Chihuahuan Desert Province (ChDP), 4 in the Baja California Province (BCP), 3 in the Balsas Basin Province (BBP), 2 in the Sierra Madre del Sur Province (SMSP) and 1 in the Chiapas Highlands Province (ChHP). Moreover, taxa of marine tardigrades were recorded from two of the eight marine ecoregions: The Northern Gulf of Mexico (NGM) (2) and the Caribbean Sea (CS) (10). Only two biogeographic provinces (Sonora and Baja California) and 6 marine ecoregions remain unexplored (Figure 1).

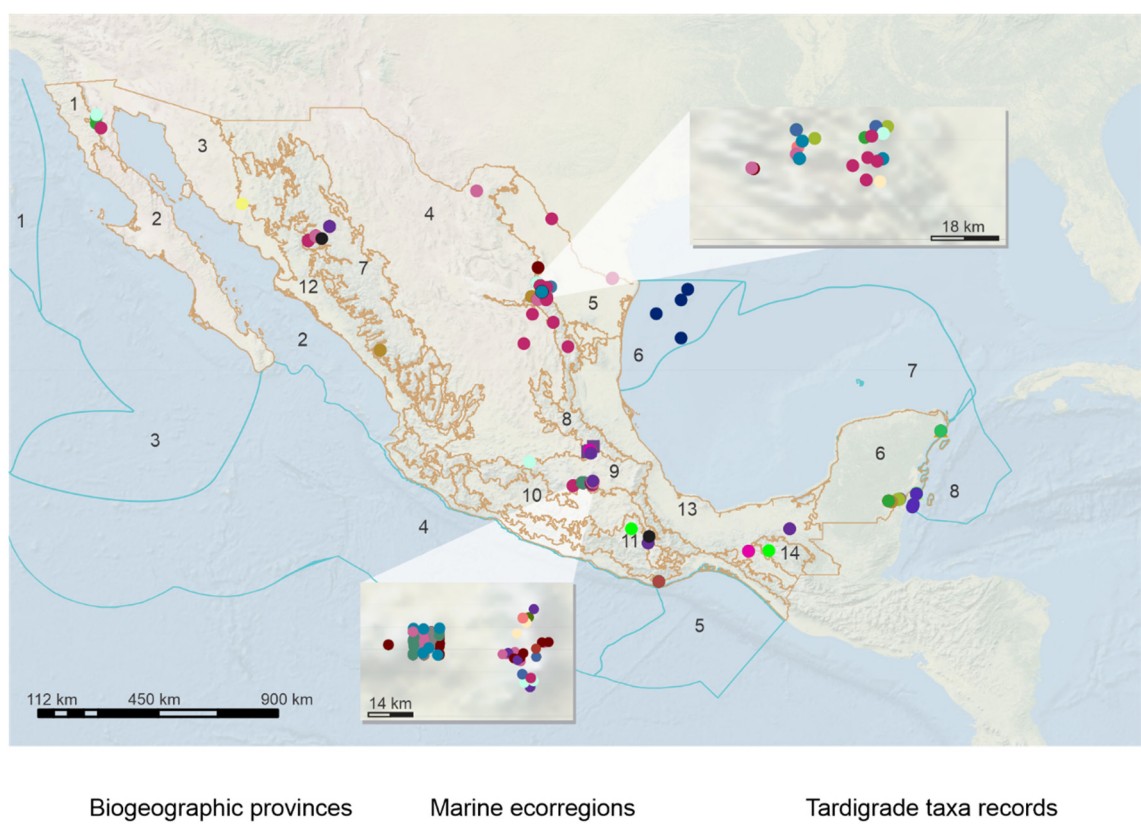

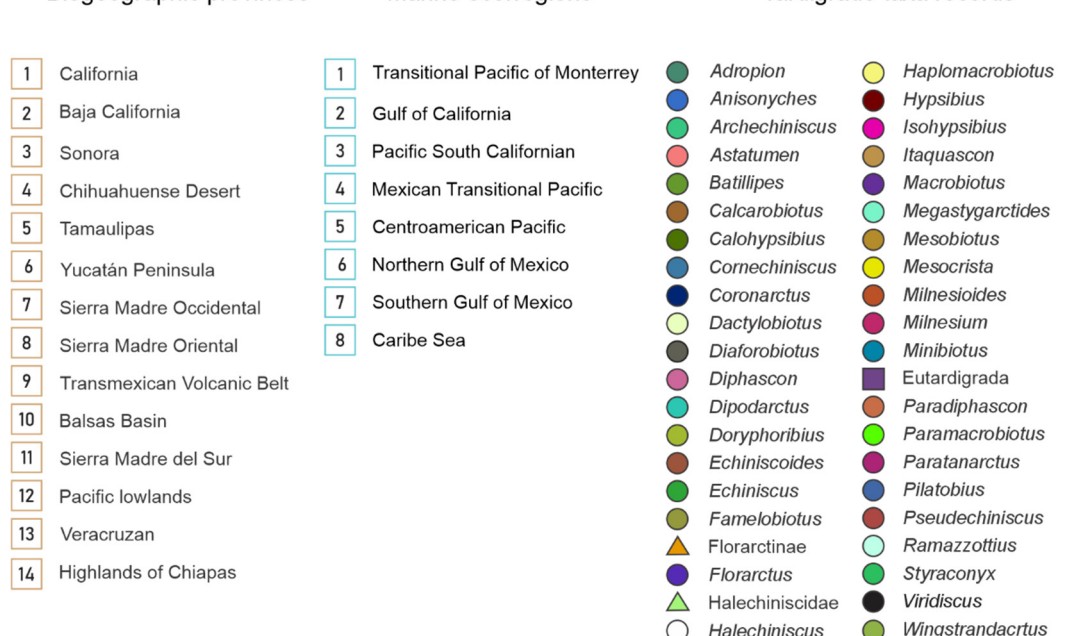

**Figure 1.** Map of tardigrade records in Mexico. Fourteen biogeographic provinces [59] and eight marine ecoregions are shown [60]. The colors indicate the taxa recorded, while the different icons symbolize the taxonomic categories: square order, triangle family, and circle genus.

Taxa richness is distributed similarly to records: 33 taxa from TVBP, 25 in SMOrP, 17 in YP, 16 in SMOcP, 11 in TP, 11 in VP, 5 in PLP, 4 in ChDP, 4 in BCP, 3 in BBP, 2 in SMSP and 1 in ChHP; while in the marine ecoregions we find 9 taxa in NGM and 2 in CS.

The list of recorded Mexican tardigrades and the integration of their national and global distribution is shown in Table 1 [25,34–49,87,88,94–101]. This information allows us to recognize that 58 taxa are present in Mexico, and display a broad distribution in at

least three biogeographic regions (e.g., *Adropion scoticum*, Figure 2a,b; *Diphascon pingue*, Figure 2f; *Paramacrobiotus richtersi*, *Wingstrandarctus corallinus*, and *Viridiscus viridis*), five taxa have records from only two regions (Paleotropical-Nearctic, Palearctic-Neotropical, and Australian-Nearctic; e.g., *Calcarobiotus* cf. *polygonatus* and *Mesobiotus diffusus*; Table 1), 33 have only been recorded in America from both Neotropical and Nearctic regions (e.g., *Diphascon mitrense*; Figure 2c,d,g; *Doryphoribius dawkinsi*, *Doryphoribius quadrituberculatus*, *Macrobiotus kazmierskii*, and *Minibiotus sidereus*, Figure 2j; Table 1), and 17 have only been recorded near of the type locality in Mexico (e.g., *Minibiotus citlalium*, Figure 2e,h).

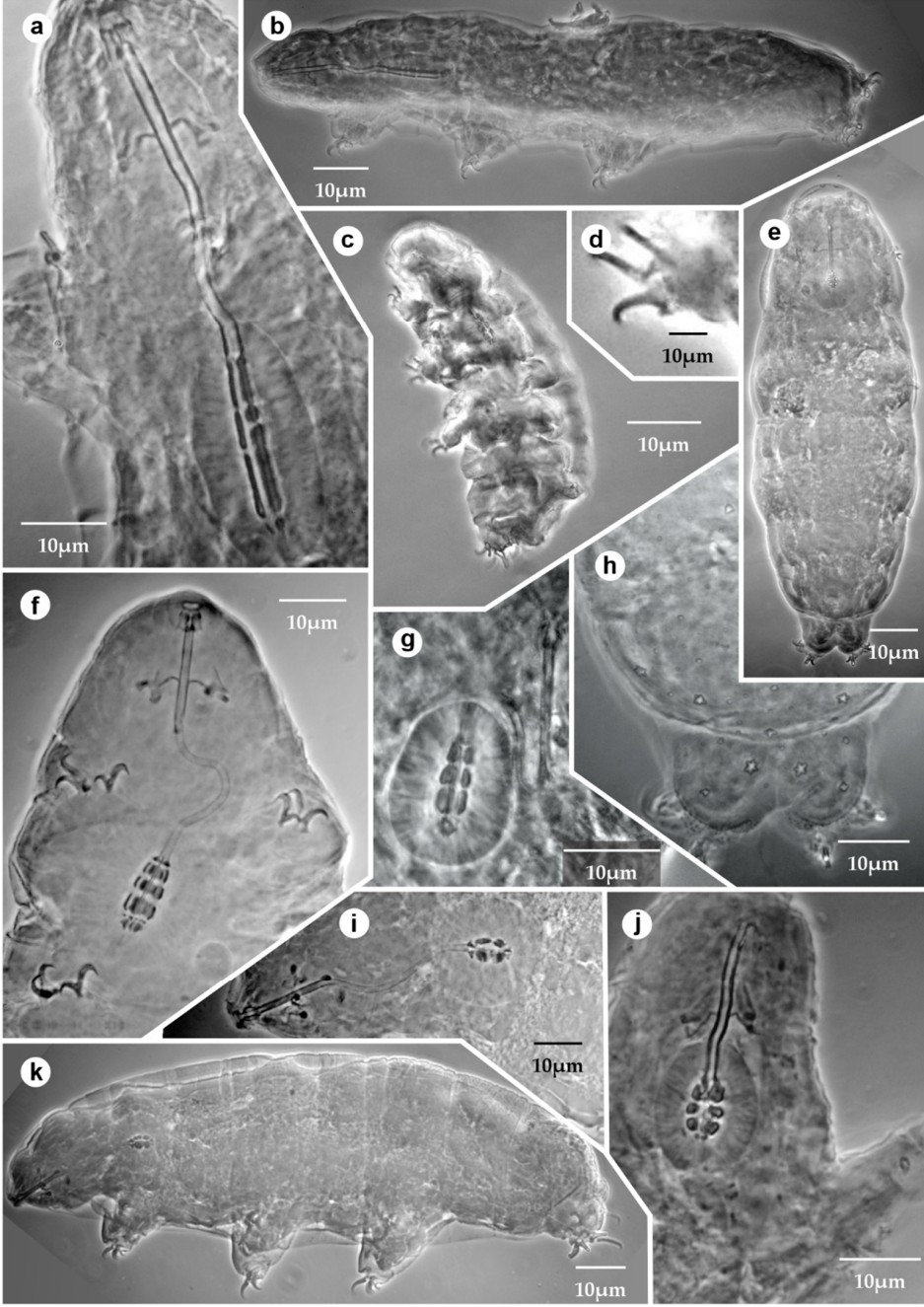

**Figure 2.** Micrographs with phase contrast microscopy of some species of tardigrades present in Mexico. Species widely distributed: (**a,b**) *Adropion scoticum* and (**f**) *Diphascon pingue*; American species: (**c,d,g**) *Diphascon mitrense*, (**i,k**) *Pilatobius nodulosus* and (**j**) *Minibiotus sidereus*; with records only on the type locality: (**e,h**) *Minibiotus citlalium*.

After this revision, the known distribution of six taxa changed (Table 1); these data come from theses, conference proceedings and conference posters. *Calcarobiotus polygonatus* had only been recorded at the type locality in Tanzania, in Mexico this species was present in Chiapas [39]; *Mesobiotus diffusus* had been recorded at several localities in Africa, in Mexico the species was collected from mosses in Coahuila and Mexico State [39,86]; the genus *Milnesioides* sp. had only been recorded in Australia, but in Mexico, this genus was also is recorded in Mexico City [35]; *Milnesium barbadosense* had only been recorded in the type locality in Barbados, in Mexico, this species was also is recorded in Nuevo Leon and Tamaulipas [25,49]; *Paramacrobiotus klimenkii* had only been recorded at the type locality in Ukraine, but in Mexico, this species also is recorded in Nuevo Leon [39]; the genus *Wingstrandarctus* had only been recorded in the Pacific and the Atlantic Ocean, in Mexico this genus was also is recorded in Quintana Roo (Caribbean Sea) [37].

### 3.3. Estimating Richness

#### 3.3.1. Ratio of Known to Unknown Species

The Mexican and global diversity data for the 15 selected taxa for the diversity estimation of Tardigrada are shown in Table 2. Based on the percentage of known species of these 15 groups, the current [21] and estimated diversity data of tardigrades [50], it the number of species in Mexico was calculated. We excluded the values of the taxa that quantified fewer than 75 species, which is the number of current tardigrades records in Mexico and obtained in this work. When the current diversity value of tardigrades was used (1380 spp.) we could find between 77 and 262 Tardigrada species, and between 126 and 505 considering the value of estimated diversity (2654 spp.) Considering the percentage of diversity present in Mexico obtained by Martínez-Meyers et al. [24], it is estimated that between 117 and 2265 species could have occurred in the country (Table 2).

**Table 2.** Ratio of known to unknown species of 13 phyla in the world, species recorded, percentage of occurrence, and the number of expected tardigrades species in Mexico. Values were obtained by extrapolating the percentage of occurrence to the number of tardigrades species in the world, of each group. The numbers in bold indicate estimated species values >75, that is, the number of current tardigrades records in Mexico obtained in this work.

| Taxa | Known Species in the World | Known Species in Mexico | Percentage of Known Species in Mexico (%) | Number of Expected Tardigrades Species | |
|---|---|---|---|---|---|
| | | | | Actual Richness * | Estimated Richness ++ |
| Plants | 390,900 | 23,314 | 5.96 | 82 | 158 |
| Mosses | 12,754 | 984 | 7.71 | 106 | 205 |
| Birds | 20,034 | 1115 | 5.56 | 77 | 148 |
| Reptiles | 10,970 | 864 | 7.87 | 109 | 209 |
| Mammals | 5629 | 564 | 10.01 | 138 | 266 |
| Fishes | 34,200 | 2763 | 8.07 | 111 | 214 |
| Insects | 955,025 | 69,163 | 7.24 | 100 | 192 |
| Arthropods | 1,013,825 | 80,000 | 7.89 | 109 | 209 |
| Protura | 804 | 17 | 2.11 | **29** | **56** |
| Diplura | 1008 | 48 | 4.76 | **66** | 126 |
| Copepods | 11,500 | 62 | 0.53 | **7** | **14** |
| Onychophorans | 186 | 3 | 1.61 | **22** | **43** |
| Nematodes | 8375 | 402 | 4.8 | **66** | 127 |
| Rotifers | 3000 | 184 | 6.13 | 85 | 163 |
| Cladocerans | 568 | 108 | 19.01 | 262 | 505 |
| Martínez-Meyer et al. [24] ° | | | 8.5 | 117 | 226 |
| Mean | | | 7.27 | 91 | 176 |

* Number of expected tardigrades species based on current richness (1380 species). ++ Number of expected tardigrades species based on estimated richness (2654 species). ° Percentage of diversity obtained for Mexico based on a list of diverse biological groups, where about 94,000 species were documented.

### 3.3.2. Accumulation Curves

The estimation based on the accumulation curves of both genus and species were adjusted to the Clench model ($R^2$ = 0.99; Figure 3, Tables 3 and 4). According to this, 78% of the genera and 39% of the species that could be found in Mexico have been effectively recorded, and around 572 samples would be needed to record 95% of the tardigrade genus and 2714 samples to complete 95% of the species in Mexico; according to the applied model, 292 species and 44 genera would be expected (Figure 3). The slope at the end of the curve at the genus level is lower than 0.1, which indicates that the inventory can be considered sufficiently reliable ($m_{genera}$ = 0.089). Although it is still incomplete, the function supports that 22% of the genera would remain undiscovered. The slope at the end of the curve at a species level is greater than 0.1, which indicates that the inventory is incomplete and unreliable ($m_{species}$ = 0.78) as the curve is far from reaching the asymptote, and 61% of the species have yet to be discovered.

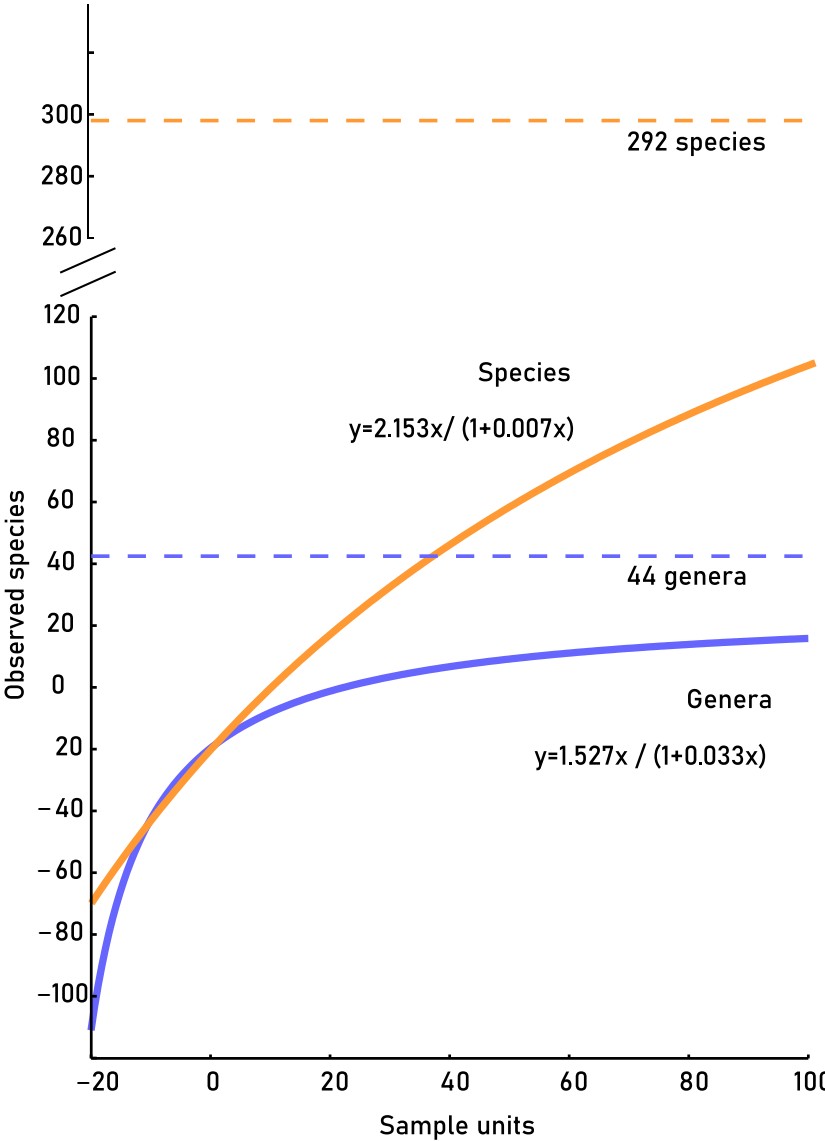

**Figure 3.** Species accumulation curves by genus (blue line) and by species (orange line). The X-axis shows the sampling effort (each geographical locality corresponding to a collection event was taken as a sampling unit). The Y-axis represents the number of species retrieved for each given sampling level. The solid lines represent the Clench function fitted to the curve. The dotted line indicates the asymptote predicted by the model.

**Table 3.** Summary of the results obtained with the species richness estimation methods analyzed in this work. The shaded rows indicate each method. We use two global richness values for the ratio of known to unknown species.

| The Ratio of Known to Unknown Species | | | |
|---|---|---|---|
| | Global richness | Range of species estimated | Mean of species estimated |
| Known richness | 1380 [a] | 77–262 | 91 |
| Estimated richness | 2654 [b] | 126–505 | 176 |
| **Accumulation curve** | | | |
| | Effectively record | Samples needed [c] | Estimated |
| Genus | 78% | 572 | 44 |
| Species | 39% | 2714 | 292 |

This range was obtained by calculating the proportion of species present in Mexico from 15 eukaryotic groups but excluding the values of the taxa that quantified less than 75 species, which is the number of current tardigrades records in Mexico obtained in this work (vascular plants, mosses, birds, reptiles, mammals, fishes, insects, arthropods, copepods, onychophorans, nematodes, rotifers, and cladocerans). [a] Current richness obtained from the Actual Checklist [21]. [b] Estimated richness obtained by Bartels et al. [50]. [c] Samples needed to complete 95% of the tardigrade fauna in Mexico.

**Table 4.** Statistics were obtained in the accumulation curve method from the Clench model. $R^2$, coefficient of determination; **a** and **b**, model constants; **m** slopes at the end of the curve; the proportion of recorded taxa, missing samples to complete 95% of fauna; and estimated richness.

| | **GENUS** | **SPECIES** |
|---|---|---|
| $R^2$ | 0.99 | 0.99 |
| **a** | 1.527 | 2.153 |
| **b** | 0.033 | 0.007 |
| **m** | 0.089 | 0.78 |
| Recorded proportion | 78% | 39% |
| Missing samples | 572 | 2714 |
| Estimated richness | 44 | 292 |

*3.4. Taxonomic Effort*

When relating the number of described species and the number of authors describing the species, we observed that the species known so far are the result of the effort of 320 authors. The histogram (Figure 4) shows an increase in the number of descriptions over time; however, there are ups and downs in the number of descriptions, one of the first increases occurred in the decade of 1904, when 57 species were described by two authors, this increase strongly was diminished by the World War I, two decades later, in which two authors described only two species. In 1924 (27 species, 5 authors), 1934 (27 species, 5 authors), and 1939 (24 species, 7 authors) there were a slight recovery in the number of descriptions; however, again a decrease was observed in the period after World War II, as in 1944 eight authors described 13 species. From the following decade (1949), there was a steady increase in the number of descriptions until 2009, when there was a decrease in the number of descriptions (88 compared to 121 in 2004) but a significant increase in the number of authors involved in this activity (63 compared to 39 in the previous decade). It is worth mentioning that the number of authors is always increasing. Likewise, in the years 2009 and since 2014 to the present, it was observed that the number of authors (Figure 4, orange bar) exceeded the average number of descriptions in the same period (Figure 4, blue bar), unlike the previous decades where the number of authors never exceeds the average number of species descriptions. In Mexico, only ten species have been described (Figure 4). After the description of the first species in 1948, no new taxa were recorded in Mexico until 2006.

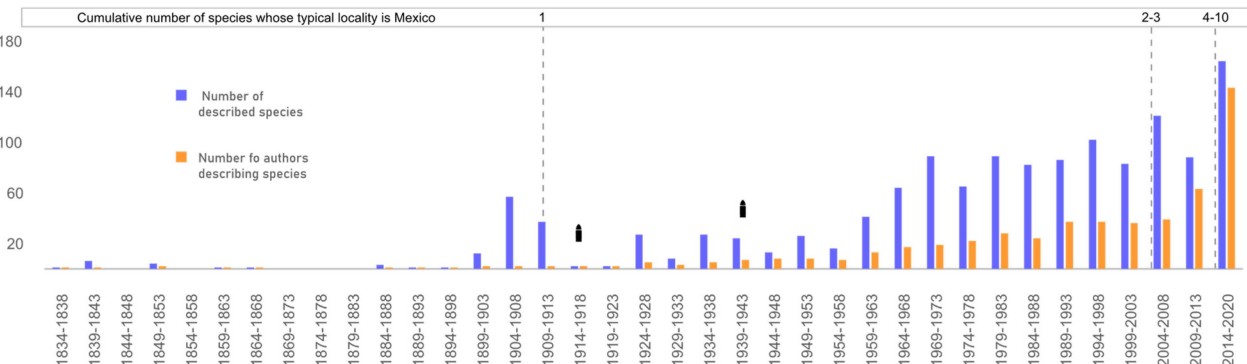

**Figure 4.** Tardigrade species described by decade from 1830 to 2020. Blue bars correspond to the number of described species worldwide. Orange bars indicate the number of authors describing species over time. At the top of the chart are the cumulative number of species whose type locality is Mexico, so far 10. The black bars represent the World War I and II.

By calculating the number of years when that the names of taxonomic authorities were recorded in the literature, we determined that one author actively describes species on average every six years (6.3 ± 0.58); however, more than half of the authors (60.31%) appeared to describe species for only one year. One author describes seven species (7.5 ± 0.99), but 44.37% of them only describe one, 14.37% describe two, and 7.5% described three species.

## 4. Discussion

Based on our strategy for the study of tardigrade diversity in Mexico, we collected 474 records for 105 taxa, belonging to 42 genera and 75 species, distributed in 12 of the 14 biogeographical provinces of Mexico [59]. With these data, it is possible to recognize that the Mexican tardigrade fauna is made up of a greater proportion of species occurring in more than three biogeographical regions of the world, followed by species only reported in the Americas and, to a lesser extent, species that have only been recorded in Mexico. Diversity estimation methods predict that more than 290 species could be found in Mexico, and 2714 samples would be required to find them.

List of the Mexican tardigrades. The study of the tardigrade fauna in Mexico began in 1911 with the work of Heinis [26] who reported four species, one new species and two genera from samples of moss and lichen from Oaxaca. Since then, to date, only two lists of the group have been published [25,33]. The first corresponds to that by Kaczmarek et al. [33] in 2011, who provided the first complete list of species documented over a century (1911–2011), which includes four new records and quantifies a total of 41 species. The second one is of Moreno-Talamantes et al. [25] in 2019, where 55 species were recognized over eight years, including 11 new records and three previously described species [39,40,45]. Although numerous records were generated in the period 2011–2019 in non-specialized literature, both of genera [34–37,42,44,87] and species [41,88], these were not included in the latter listing, nor in the calculations of national diversity presented in later works [43,47]. From 2019 onwards the diversity of tardigrades increased notably in Mexico, due to the description of new species [46–49] and the recognition of new records at the specific and generic level [38,43,47,49]. For example, in 2020 alone, 61 species were recognized [47] and in 2021 84 species [43]. However, these works did not provide a list of taxa in support of their calculations on the number of species.

According to our results of a detailed literature review on Mexican tardigrade fauna, 105 taxa are recognized for the country, of which 75 were "identified at species level", 27 are assigned at genus, two to a subfamily, and one to order levels. Of the 75 taxa identified at a species level, only for 62 of them were the authors able to support the specific assignment, with qualitative and quantitative characteristics, while in 13 (*Dipodarctus* cf. *subterraneus*, *Echiniscus* cf. *tamus*, *Halechiniscus* cf. *perfectus*, *Pseudechiniscus* cf. *juanitae*, *Calcarobiotus* cf. *polygonatus*, *Calohypsibius* cf. *ornatus*, *Hypsibius* cf. *convergens*, *Hypsibius* cf. *microps*, *Hypsibius*

cf. *pallidus*, *Macrobiotus* cf. *acadianus*, *Minibiotus* cf. *intermedius*, *Paramacrobiotus* cf. *klymenki* and *Ramazzottius* cf. *oberhaeuseri*), due to lack of congruence in the quantitative measures, it was only possible to assign with certain probability their belonging to the identified species, and therefore they constitute putative species. Some records at the genus level, some are specimens which lack either body morphological attributes or eggs, hindering assignation to a specific level. Several of them are potentially new species represented by unique individuals that have contrasting characters with the species recognized for those genera [37]. The record at the subfamily level (Florarctinae) [37] was supported by specimens that were subsequently identified at the species level [46], and the record at the family level (Halechiniscidae) comes from a single specimen that presents unique characteristics, suggesting the occurrence of a new genus.

Based on this list, we recognized that 28.3% of the taxa have not been determined at a specific level, and in others, it has not been possible to delimit new taxa based solely on morphological characteristics, which is evidence of the absence of an integrative strategy for the study of tardigrades diversity. The use of integrative methods that include different types of observation of morphological characters, and molecular markers focused on the delimitation of taxa, has been documented in other works for the phylum [18,19,103–105]. The use of a multidisciplinary approach that takes advantage of the complementarity between different techniques of analysis of morphological [106] and genetic variation analyses [107] will allow clarifying the status of the taxa in Mexico and will provide more robust identifications and hypotheses testing of species.

An example of the urgent need to use integrative methods of species description is the number of DNA sequences deposited in the GeneBank database for described tardigrade species so far (more than 1300). It is observed that molecular information can be found for only 294 species, of which 123 species include fragments of the mtDNA Cytochrome Oxidase gene, four species have fragments of the 16S rRNA, 163 species have fragments of the 18S rRNA gene, 135 species have fragments of the 28S rRNA gene, and 156 species have ITS sequences [22]. However, some of these sequences have been obtained for purposes other than taxonomy, so fewer than 294 species have been described with molecular information sources.

Distribution patterns of the Mexican tardigrades. Our records support the presence of tardigrades in 17 of the 32 states of the country. Nuevo León, Mexico State and Chihuahua, were the states with the most records (23, 20 and 14 respectively). The Mexican biogeographical provinces with the greatest diversity of tardigrades, both at species and genus level, were the Transmexican Volcanic Belt (TVBP), the Sierra Madre Oriental (SMOrP), and the Sierra Madre Occidental (SMOcP), which have been previously identified as particularly species-rich regions [108].

Looking at the distribution pattern of tardigrade taxa records in the biogeographical provinces, it is evident that exploration of the phylum has not been uniform and there are many unstudied areas (Figure 1). In the TVBP, records are concentrated in the east of the province (Popocatépetl and Iztaccíhuatl volcanoes); in the SMOrP and SMOcP, records are clustered in the northern region (Nuevo León and Tamaulipas, and Chihuahua respectively); in the Yucatán Peninsula (YPP), all records are located in the northeastern region of the province (Quintana Roo); in the Veracruzan province (VP) the records are concentrated in the southwestern region (Chiapas) and in the Tamaulipan province (TP) the records are concentrated in the northern region (Nuevo León, Tamaulipas, and Coahuila). There are also provinces with few records, such as Sierra Madre del Sur (SMSP), Pacific Lowlands (PLP), and Balsas Basin (BBP), which had less than 5 records per province. These areas have the potential to considerably increase the inventory of the Mexican tardigrade fauna, since in other groups such as vertebrates, vascular plants, and arthropods, a great richness of species is recorded in the states of Oaxaca, Veracruz, and Chiapas [109].

Of the 105 recognized taxa in Mexico, 54.72% have a wide distribution (occurring in more than three biogeographical regions of the world), 14.14% have records in two biogeographical regions and 31.12% have only been documented in American provinces.

The latter corresponds to 30 taxa, of which only 11 species distribution patterns could be recognized based on their current records. One of these patterns is the Nearctic, found in *Macrobiotus anemone* and *Pilatobius nodulosus*, which have records from Canada, USA, and Mexico. Another pattern was Neotropical, found in the species *Doryphoribius quadrituberculatus*, *Diphascon mitrense*, *Echiniscus kofordi*, *Macrobiotus alvaroi*, *Macrobiotus kazmierskii*, *Milnesium barbadosense*, and *Minibiotus sidereus*, with records from the Mexican Transition Zone (MTZ) [110,111], Argentina, Chile, Costa Rica, Ecuador, and Barbados. Two species also extend their distribution from the MTZ to the Nearctic and Neotropical regions: *Doryphoribius dawkinsi*, found in the USA, Mexico, and Costa Rica, and *Diphascon chilenense*, found in Canada, USA, Mexico, and Chile.

Some tardigrades with Neotropical records have distribution similarities with other groups of animals and plants. *Macrobiotus alvaroi* was detected in the Mesoamerican (Chiapas) and Pacific (Costa Rica) domains of the Brazilian subregion; a similar distribution has been recognized in the coleopteran *Deltochilum acropyge* [110], whose distribution was attributed to the Neotropical cenocron with minimal penetration; the taxa supporting this pattern have a South American origin and have entered into the large rainforest patches of the southeast, in Chiapas and Campeche [112].

Other examples are *Echiniscus kofordi* and *Milnesium barbadosense*, species that are distributed in different subregions, from the Neotropic to the Nearctic, and show similar distribution patterns with some moss species. *Echiniscus kofordi* is found in the Brazilian (South American transition zone domain in Ecuador and Mesoamerican domain in Chiapas) and Allhegani (Louisiana, Alabama, and Florida) subregions, a distribution that is consistent with the moss species *Atractylocarpus flagellaceus*, representative of the Mesoamerican element of the moss flora, which is constituted by 174 species that extend mostly from Mexico to South America, but in some cases, they present a wider distribution towards the southeastern USA [113,114].

*Milnesium barbadosense* is found in the Antillean (Barbados) and Brazilian subregions (Mesoamerican Domain in Chiapas and the Pacific Domain in Colombia), a distribution that is in agreement with the moss species *Breutelia jamaicensis*, *Calyptothecium duplicatum*, and *Helicophyllum torquatum*, representatives of the Caribbe element, which is characterized by species shared by North and South America and whose presence in the Antilles is interpreted as migration from the continental masses [114]. To confirm any of the patterns described above, more records are needed to test these hypotheses.

Based on our exhaustive literature search, 14 new records were found, mainly from sources such as scientific outreach journals, theses, conference proceedings, and posters, which could be considered "less rigorous", since these records are not subjected to the same scrutiny by specialized taxonomists, as those records are published in a scientific journals; therefore, it is necessary to perform rigorous and appropriated studies to confirm these 14 additional records.

Furthermore, the knowledge of Mexican marine tardigrades is very recent. The first record was obtained in 2019, in fact, of the total of analyzed works, 12% corresponds to the study of marine tardigrades. This is a global phenomenon, as marine tardigrades are less well known than limnoterrestrial ones [50], which comprise approximately 15% of the tardigrade fauna [21]. In Mexico, marine tardigrades have been collected at depths ranging from 0.2 m in the littoral zone to 2847 m depth in the bathypelagic zone [38], while in the rest of the world they have been found up to 4170 m [115]; therefore, in Mexico, there remain coast and deep ocean areas missing for exploration, such as the abyssal zones present in the Gulf of Mexico, Caribbean and Atlantic Ocean [116].

One of the most interesting characteristics of tardigrades is their adaptation ability to hostile environments; these traits have supported the idea that tardigrade species have been considered largely ubiquitous and cosmopolitan. Early alpha taxonomy studies using morphological characteristics as the main form of identification have led to the formation of species groups or complexes, e.g., *Echiniscus arctomys*, *Macrobiotus hufelandi*, *Milnesium tardigradum*, *Paramacrobiotus richtersi*, etc. However, results from recent integra-

tive studies indicate that tardigrade species may not be as widely distributed as previously thought [117–123]. This idea is supported by distribution patterns reported in other metazoans [124–129].

Estimating richness. In 2014, a series of 29 publications were made to analyze the Mexican biodiversity, in which 56 biological groups quantifying the number of species occurring in the country, as well-developed predictions on how many more species remained to be found [24]. Our calculation of potential diversity with the ratio of known to unknown species indicate that 6.1% of the global diversity of the 15 eukaryotic groups analyzed are found in Mexico, a very similar value to that found by Martínez-Meyer et al. [24] of 8.5% (although the biological groups analyzed in both studies are not the same). Martinez-Meyer et al. [24] analyzed microinvertebrate groups, such as springtails [130], proturans [77], and diplurans [78], leaving aside some groups of invertebrates registered in the country (tardigrades among them they). Thus, until now, there was no formal calculation of the current and estimated richness of the Tardigrada phylum in the country.

As in all richness estimation methods, the ratio of known to unknown species method, used in the present study, has its drawbacks. In the case of tardigrades, a very wide range of richness value was obtained because groups with very little presence in Mexico were used (proturans, copepods, onychophorans, and others highly abundant, such as cladocerans). These groups included, as tardigrades are more similar in habitats and lifestyles, to Protura, Diplura, Rotifera, free-living nematodes, Collembola, and mites; therefore, more real richness ranges were expected. Another important point is that extrapolating diversity from one taxon to another may be considered guesswork.

The accumulation curve method determined that for both genera and species the inventories are incomplete because 22% of the genera and 61% of the species have yet to be discovered; however, estimates of the number of species derived in this way often lack associated margins of error, making it impossible to objectively assess their accuracy [131,132]. Moreover, for genera level, the inventory is reliable and for species it is unreliable. In other studies of invertebrates [133–135], the reliability of the inventories depends on the study group, their vagility, and habitat characteristics. An example of this are the families Araneidae, Thomisidae, and Salticidae (Arachnida); in the first two families, it is possible to obtain more complete inventories in plots of 1 km$^2$ [136], while in the family Salticidae the inventories obtained in an area of equal size are far from complete, due to the abundance of rare species [89]. In the case of the tardigrades, it is known that the species of the genus *Macrobiotus* are found in a wide range of habitats and hosts, being one of the most abundant genera in bryophytes [137,138], while other taxa seem to be restricted to a range of habitats and hosts. For instance, in the Iztaccíhuatl volcano in Mexico, the genera *Adropion* and *Pilatobius*, are apparently restricted to the *Abies* and *Pinus* forest, and the *Hypsibiidae* family are only present in the Mixed forest, *Abies religiosa* and *Pinus hartwegii* forest [47]. Therefore, to get complete the inventory, the sampling effort must be directed to a wide spectrum of environmental conditions, in addition to being directed to fulfill the objectives of the study.

Taxonomic effort. In the period from 2006 through to 2021, ten tardigrade species were described in Mexico, which correspond to 0.66 species per year. The descriptions of these species are found in seven publications and six of them were made by Mexican authors [25,45–49], meaning that Mexico contributes to 0.44% of the authors and 0.43% of the tardigrades known so far. This situation can be observed in other countries, for example, in Colombia, a mega-diverse country, from 2014 to 2020 nine tardigrade species have been described, which corresponds to 0.67% of the global tardigrade diversity, contributing to 1.25% of the species authors [139–142]. Unfortunately, in other countries within the region, such as Trinidad and Tobago, Belize, Guatemala, El Salvador, and Honduras, among many others, tardigrade species remain undiscovered.

The number of tardigrade taxonomists in Mexico and Colombia is an example that in the last 10 years, more taxonomists have joined the study of tardigrades in their countries of origin. Although fortunate, the study of tardigrades from this geographic region is in the alpha phase, which implies intensive collecting, detailed observation, description,

and species naming [143]. Therefore, continuity in the training of specialized taxonomists involved in the study of tardigrades is imperative, as well as the design of taxonomic work within a comprehensive research project.

Worldwide, the diversity of tardigrade species described by 10 prolific authors ranges between 50 and 204 described species per author (Pilato 204 species, Binda 111, Lisi 92, Kaczmarek 84, Michalczyk 74, Maucci 58, Murray 58, Mihelčič 57, Iharos 56 and Dastych 54 species), and in total, these authors have described 848 species (63.37%) of those known to date. Such a scenario in which prolific authors describe the bulk of species can be observed in numerous groups. A clear example is in bees, on which almost 19,508 known species in the world, and the 10 most prolific authors have described from 417 to 3394 species, and in total, these authors have described 9435 species (48.36%); the author who described the most species (3394 species) contributed more than 17% of the species so far known [144]. Comparing the number of authors who describe bees and tardigrades, it is evident that there are fewer descriptions in Tardigrada; however, the percentage described by prolific authors is considerably higher.

Finally, it is important to mention that another factor that has maintained a low rate of species description is the lack of inclusion of methods of analysis for their delimitation, particularly in taxonomically complex groups, which is reflected in the fact that 25.7% of the species recorded in Mexico are identified as "*confer*" (cf.); the inclusion of different methods of analysis of morphological variation, character observation techniques, and molecular data analysis in an integrative context will allow the recognition, delimitation, and generation of more robust hypotheses of the documented species.

## 5. Conclusions

Records of the tardigrade fauna in Mexico have been based on morphological characters such as size and shape of the claws, organization of the buccopharyngeal apparatus, cuticular patterns, and egg morphology. However, these structures present intra-specific variation that has not been quantified with more precise methods; it is, therefore, necessary to use integrative methods using different sources of information, such as ecological data, molecular data from nuclear and mitochondrial DNA, and morphological characters. These would be invaluable to future studies.

From the information obtained by a literature review, 105 taxa were recorded, belonging to 42 genera and 75 species. According to the bibliographic source, 31.85% of the records were obtained from indexed journals, 65.6% from theses, 1.89% from conference proceedings, and 0.63% from scientific outreach journals, so many of the records reported need confirmation. The best-sampled provinces in the country were TVBP, SMOrP, and SMOcP and the least explored were SMSP, PLP, and BBP, although in all provinces, records are concentrated in one or two states, leaving much of the territory to be explored. Moreover, two biogeographic provinces and six marine ecoregions remain unexplored, which in turn should stimulate future research and Tardigrada studies in the country. Another factor to evaluate in terms of tardigrade diversity is their ocean depth distribution since in Mexico there are still coast and deeper areas to be explored, such as the abyssal zones that are present in the Gulf of Mexico, the Caribbean, and the Atlantic Ocean.

After this revision, the known distribution of six taxa was expanded as sampling efforts intensified in Mexico and other regions of the world. This pattern of spread distribution will become increasingly common. Most of the species found in the country have a wide distribution, except for species that have been recorded only from the American continent. Likewise, many species (14) have only been recorded in Mexico, so it can be said that they are temporarily limited to the type locality until new records are found that confirm their presence in other places.

For the actual detection of missing species in future studies, it is necessary to carry out systematic sampling in different types of vegetation and habitats in poorly explored areas, as well as to apply integrative methods that facilitate the delimitation of species. We also want to attract the attention of scholars of this group and encourage them to publish

their results. In Mexico, this phylum requires a lot of studies, and any contribution adds to the knowledge of the group; in addition, the exercise that involves the presentation of a manuscript for its publication adds to the taxonomic and writing experience, which is essential in the formation of specialists in the group.

**Author Contributions:** Conceptualization, F.A.-T., E.A.R.; methodology, J.G.-R., M.C.-E., A.D.-C., E.A.R., F.A.-T.; software, J.G.-R. and M.C.-E.; validation, F.A.-T., J.G.-R.; formal analysis, J.G.-R., M.C.-E. and A.D.-C.; investigation, J.G.-R., M.C.-E., A.D.-C., F.A.-T.; resources, J.G.-R., M.C.-E., A.D.-C. and F.A.-T.; data curation, J.G.-R., A.D.-C.; writing—original draft preparation, J.G.-R., F.A.-T.; writing—review and editing, J.G.-R., J.J.F.-M., C.F.V.-M., E.A.R., F.A.-T.; visualization, J.G.-R. and F.A.-T.; supervision, C.F.V.-M., E.A.R., F.A.-T.; project administration, F.A.-T., E.A.R. and J.J.F.-M.; funding acquisition, C.F.V.-M., J.J.F.-M., E.A.R. and F.A.-T. All authors have read and agreed to the published version of the manuscript.

**Funding:** This research was funded by the SIP-IPN 20200591, SIP-IPN SIP20195761, SIP-IPN 20210142 (E.A.R.), PAPIIT-UNAM IA201720, IA203122 and CONACyT Fronteras de la Ciencia (139030) (F.A.-T.).

**Institutional Review Board Statement:** Not applicable.

**Informed Consent Statement:** Not applicable.

**Data Availability Statement:** Not applicable.

**Acknowledgments:** We are grateful to four anonymous reviewers for their comments. We also thank Pedro Mercado Ruaro from the Laboratorio de Botánica Estructural, Instituto de Biología-UNAM, for phase contrast photographs.

**Conflicts of Interest:** The funders had no role in the design of the study; in the collection, analyses, or interpretation of data; in the writing of the manuscript, or in the decision to publish the results.

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
