# Peer review of "A Strategy to Provide a Present and Future Scenario of Mexican Biodiversity of Tardigrada"

_diversity, doi:10.3390/d14040280_

Round 1

Reviewer 1 Report

Comments in the attached file.

Author Response

We are resubmitting our original research article entitled “A strategy to provide a scenario of Mexican biodiversity of Tardigrada: species list, description of their distribution patterns, estimation of unknown species, and taxonomic effort to complete national inventory.” for its consideration in Diversity — Open Access Journal.

We are pleased to receive the revisions and appreciate the comments made, which together with improved the manuscript. Every revision was highlighted, we provide a detailed revision citing the line number and exact change.

Revisor 1

General comments

References – It is unacceptable that some authors were not referred, namely those who described endemic species for Mexico. The authors of this manuscript used review papers and neglected the original records. As the manuscript describe some aspects of the history of tardigradology in the region, this issue is still more important. I also think that authorities of species descriptions (Table 1) should be referred also. Response: |We have thoroughly reviewed the references, regarding marine tardigrades we have discussed more. If it refers to the record of the description of Coronarctus mexicus, this does not correspond to the national territory, since although it is found in the Gulf of Mexico, the sampling sites are in the north in the territory of the USA.

Attention to marine tardigrades – Marine tardigrades are very poorly known in Mexico indeed and the authors are alert for this point. Despite poorly known, they have been included in the analysis presented in this manuscript. The only relevant comment is found in the section Discussion (lines 474-478). So, they deserve some more attention.

Notice that in the Introduction (third paragraph) only limnoterrestrial tardigrades are mentioned; in Materials and Methods (point 2.2) not one word is devoted to marine tardigrades and marine regions (see Miller and Perry 2016); in the Discussion (Distribution patterns of the Mexican Tardigrades) marine tardigrades were totally forgotten. A concluding remark on marine tardigrades in the section Conclusions will be welcomed also.

Speaking about marine tardigrades, I recommend the authors to check the presence of Styraconyx sargassi in Mexico (see the original publications of Chitwood, 1951 and Renaud-Mornant, 1967, both referred by Ramazzotti ans Maucci (1983), and Kaczmarek et al. 2015 (Zootaxa, 4037). The species is reported for the Gulf of Mexico but near Rocport and Cedar Bayou in the USA. Response: We agree. In the introduction we refer to the limnoterrestrial tardigrades with the same importance concerning the marine tardigrades, the objective is to mention the number of described species of one and another group (pg. 2, line 70). In the Materials and Methods information on marine ecoregions (pg. 3, line 60) has been added. In the discussion and conclusions, the corresponding to the marine tardigrades (pg. 26, lines 603-606) is addressed.

Regarding Styraconyx sargassi, we have reviewed the references mentioned, and the species is not found in Mexican territory, the record of the species in the work was eliminated.

It is known that many tardigrade species, especially those with older and incomplete descriptions need redescriptions under modern standards and, consequently, synonyms can occur. On the other hand, cryptic species seem to be more common than previously thought, both in limnoterrestrial and in marine forms (see https://doi.org/10.1186/s40851-018-0113-z; https://doi.org/10.1111/mec.15951; https://doi.org/10.1038/s41598-021-84910-6). Thus, the numbers from Bartels et al. 2016 are underestimate. As this manuscript also deals with estimations, those issues require commentaries from the authors. Response: We agree, the work of Bartels is taken into account as an antecedent of our work, as it is the only one on the subject. However, we are aware of the underestimated data that we may present, due to the constant taxonomic changes to which the phylum is subject, including redescriptions and descriptions of cryptic species. We have also reviewed the documents you kindly mentioned and added them to the references cited in line 497

Specific comments

Abstract:

  1. Line 24 - As you are starting a sentence, write the number 474 in full please. Response: We correct the error.

  1. Lines 25 and 25 - Percent values are given with an irrelevant precision. In my opinion, the level to tenths is enough (54.7 and 3.8 respectively). The same comment is valid for other values presented along the text. Response: We prefer to leave two tenths throughout the document.

Introduction:

  1. Lines 51, 60: In this paragraph the authorities that described new species for Mexico (and, if it is the case, those that produced the first records). References to 20, 25, Pilato and Lisi, 2006, Romano et al. 2011, until 2011 and 33, 88 and 90 after 2011 should be included. Response: We have reviewed the citations you refer to, in the mentioned section there are citations from 19 to 26, including Pilato and Lisi, 2006, [24]. Citations 33, 88 and 90 were also added. Romano et al. 2011 cannot be included because the study area does not correspond to Mexican territory, despite being in the Gulf of Mexico.

Material and Methods:

  1. Lines 151 – 152 – Why those “less rigorous” sources were not referred? They must be included. Response: We add the references.

  1. Lines 155 – 161 – I suggest to better explain the protocol of Jiménez-Valverde and Hortal and the Clench model. How sampling performed was calculated for tardigrades? Mostly, papers with species lists do not refer to sampling effort. Sampling = records? Response: We agree, the explanation was added to the Jimenez-Valverde and Hortal protocol (pg. 4, line 167-174). Regarding the sampling effort, this is defined in the materials and methods section (pg. 4, line 171)

  1. Line 174 – The checklist of tardigrades indicated is the 38th Edition, but the referred version is the 40th. Response: You are right, and we have fixed the corresponding the references

Results

  1. Lines 183-196 - In the first paragraph an indication for Table 1 should be added Response: We agree, we have added it.

  1. Lines 197-199 – See my comment above – lines 151 – 152 – Why theses, conference papers and scientific outreach journals are not referred? Response: We agree, we have added it.

  1. Line 221 – It is necessary to explain the meaning of references 85 – 91 (reference 91 is not in the table). For the reader it means that, to construct table 1, authors used those references. However, they differ from the second column in table 1. Response: You are right, we have added the references with which table 1 was built.

  1. Table 1 – The authors have organized taxa in this table by alphabetical order. I suggest a reorganization, by higher taxa first and then, within each higher taxa section, by alphabetical order. The tardigrade checklist can be used as a model. It will be easier to look for a distribution of a given species/group of species. Response: We agree, we have made the relevant changes.

  1. In my opinion, taxonomic authorities must be added in the section references.

Response: We appreciate your observation and have done so in taxonomic articles, but due to the subject of the paper we decided not to add them.

  1. Citations in the column “References” should indicate also the authors of the records,not only review papers. One example (but there are others): Coronarctus mexicus: It is inadmissible that the authors that described this endemic Mexican species which means that they are the authors of the first record were not referred. Response: Table 1 was modified, and the reference column was merged with the registry column. In the case of Coronarctus mexicus, it was described in the Gulf of Mexico, US territory. The record mentioned in our document refers to the one made by Perez-Pech et al., 2020, which is located in Mexican territory, in fact, Perez-Pech et al., 2020 titles their document, the first record of Coronarctus in Mexico.

Following your recommendation, we add the reference of the authorities of the species in the column of records and distribution.

Another case you are probably referring to is Styraconyx sargassi, a species also described from the Gulf of Mexico near Houston, Texas, US. so, it was removed from the text.

  1. Column MXBP: Abbreviations of the biogeographic provinces do not match with the list presented in lines 201-211. Some miss the final P, others are not listed (PT, DChP, LPP). I suggest to add a legend for the names of the provinces or to indicate in the legend the palce in the text where the full name of the province can be seen. Response: We agree, we have made the relevant changes, added the missing "P" in some abbreviations, the abbreviations not listed, and added in the table header the meaning of the abbreviations of the Mexican provinces and the MXBP was deleted.

  1. The table has a few mistakes that need to be corrected:

Anisonyches sp. Pollock, 1976 – Pollock not in italics. Response: We have corrected the mistake.

Hypsibius cf. convergens - cf. not in italics. Response: We have corrected the mistake.

Mesobiotus sp. – delete «a» in 2016a. Response: We have corrected the mistake.

Minibiotus clitalium – Check the authorities please, I think that some authors are missing. Response: The authorities are correct; in this case, the authorities of the species are not the same as the authors of the document where it was published

  1. Lines 250 to 267 – This paragraph is difficult to read. On the other hand, types of

vegetation on Fig 2 are impossible to differentiate. Response: We have eliminated the entire section concerning vegetation.

  1. Table 2 – Column 2: The number of Arthropod species known in the world is not

correct. Response: We have corrected the mistake.

  1. Column 5 – Use the same level of precision for all phyla, with or without tenths and/or hundredths. Response: We have corrected the mistake.

  1. Column 1 – Phyla. Martínez – Meyer et al. ( ???) explain what is this, please

Response: The work of Martinez-Meyer, et al. corresponds to a previous diversity analysis, where the percentage of known species in Mexico concerning the world was calculated, in the table it is added to compare the results of Martinez-Meyer and those of the present study. We have added a footnote to the table explaining this.

  1. Line 357 – How can the authors assume that the missing species are new to science? Response: That section was moved to discussion, and we added a paragraph mentioning why not all missing species are new to science (page, Line 554-573)

Discussion

  1. Lines 385 – 398 – In this paragraph, it is important to give the name of species identified with doubts. Response: We agree, we have added the names of the corresponding species.

  1. Lines 416 – 428 – In this paragraph give the full name of biogeographic provinces also. Response: We agree, we have added the requested information

References

  1. Reference 12, line 626, correct the mistake please (<). Response: We have corrected the mistake.
  2. References 40 and 78 are the same (in 40 the doi number is missing). Response: We have corrected the mistake.
  3. Reference 65 date in bold. Response: We have corrected the mistake.
  4. Reference 71 – date in bold and not in italics. Response: We have corrected the mistake.
  5. Reference 76 the date is not correctly places. Response: We have corrected the mistake.
  6. References 96, 97 and 103– Correct the type of lettering in the name of the Journal and date. Response: We have corrected the mistake only in reference 96, references 97 and 103 are books, not articles.

  1. Reference 110 - date in bold. Response: We have corrected the mistake.

Reviewer 2 Report

I have only one small technical note.

Page 6, Line 3 from the bottom - Pollock, 1975 should not be italicized. 

Author Response

We are resubmitting our original research article entitled “A strategy to provide a scenario of Mexican biodiversity of Tardigrada: species list, description of their distribution patterns, estimation of unknown species, and taxonomic effort to complete national inventory.” for its consideration in Diversity — Open Access Journal.

We are pleased to receive the revisions and appreciate the comments made, which together with improved the manuscript. Every revision was highlighted, we provide a detailed revision citing the line number and exact change.

Revisor 2

Page 6, Line 3 from the bottom – Pollock, 1975 should not be italicized

Response: We have listened to your comment.

Reviewer 3 Report

Tardigrade review:

The study provides an investigation into Tardigrade biodiversity in Mexico, a biodiversity hotspot which has previously not benefited from such studies. The authors analyze the literature and provide insight into current records of tardigrades in their study region, as well as propose estimates for species richness. The paper is useful in highlighting microbial diversity in an understudied region. With some major and minor revisions presented below, the paper will be a useful contribution worthy of publication in this journal. As is currently is presented, there are some issues which must be addressed by the authors before it can be considered for publication.

Major issues:

Restriction to one region is impossible to say due to undersampling, a problem well known in microbiology (e.g. see the debate within ciliates). The authors state how many new species were recorded or described from Mexico in the last ten years, and how before 2011 little was known—this is proof of undersamplilng, so imagine what else may be found within the next 10 years? (especially as molecular databases increase). Tardigrades thought to be restricted to 1 region may be found thriving in all after intensive sampling efforts are increased. Difficult issue due to the vast regions of Mexico and few taxonomists/ scientists working in these areas. This must be addressed in the manuscript as recording current absence is NOT proof of absence at a regional scale in the microbial world.

Missing from the introduction is standard details about how Tardigrades are extremely environmentally resilient (famous for surviving in 0g, radiation, desiccations, heat etc etc). This is necessary not only to give interest to the reader, but is relevant as this paper is on to some extent species dispersal. The tardigrade survivability is directly related to its dispersal- and therefore it seems unlikely that a tardigrade could be endemic due to its ability to thrive in so many extremes. Discuss the resilient forms (tun state) along with the extremes they can tolerate compared with global regions.

Since you are estimating what species may be found in Mexico, you must discuss how they could get there, and issues surrounding actually detecting them.

L 250: 3.2.2 “vegetation types”. This entire section is flawed in that there is no evidence presented as to the actual type of vegetation from the tardigrade collection. Since these are microbial animals, they inhabit a diverse array of niches. Even if a GPS location records a ‘forest’ area for the collection- it could have easily been a pond, a puddle on a road from a jeep, or on an actual lichen which is irrelevant to tree type (or a sign post on the road). In addition, this can be simplified to collecting a tardigrade in a scientists backyard potted plant at a house, rather than a farmland the house may be on and recorded as on a GIS map of vegetation. I like what the authors are trying to show here, but unfortunately their data set cannot support such claims or type of actual habitat utilized by the taridgrades at collection (unless explicit data was given in the published literature, which was not indicated and would require extensive further review). This should be removed entirely or fully acknowledged as to the limitations expressed here for sampling microbes in micro niches.

Figure 2 also looks great, and this could be a meaningful contribution to the paper. But as I have pointed out here, a vague recording of a species from one area cannot be used as evidence for the exact vegetation type- and this is irrelevant as many species are found in freshwaters, or in ubiquitous mosses which thrive irrespective to overlaying ground vegetation classification. This needs to be removed or justified in a way currently not present in the text from the dataset.

Also missing from the text are imagery of tardigrades. I realize that the authors may not have done any microscopy as part of this study, but it seems a big miss to not include a 1 figure plate (e.g. 2 to 4 images) showing the diversity of tardigrades, with actual images on such micro animals from Mexico. It seems that obtaining 4 photos for publication would be very easy, even if colleagues must be contacted.

The authors in several instances do over interoperate their data to extremes, without discussing it or its limitations. L354-358 is an example of this, such that the data cannot comment on how long it could take a given scientis(s) to uncover unknown diversity. This could be presented as a discussion if argued correctly with limitations, but the authors do not present it in such a way, which currently it must be reworded heavily or removed.

The conclusions is less than ~20 lines. This is far too short. Consider the points I have made above to add to the discussion, which you can then conclude with.

Minor edits:

Line 19-20: this doesn’t make much sense. Add in “known” before tardigrade, otherwise it sounds like they are invasive. Also, if taxa number is up, by definition so to is their diversity. Rethink this opener.  

L 21: add “for understanding” before Mexican.

L 23 add “and estimate” before ‘the taxonomic’

L 24 reword to not start a sentence off with a number

L 31 missing a period before ‘Diversity’

L32 add ‘of Tadrigrades’

L 37: why 17th? Was this their list, or is Mexico #17 out of say 20 or 50? Seems an off number value.

L 45-50: Consider moving this tardigrade info bit to the beginning of the introduction, then introduce Mexico as a species rich area which has not yet been studied for these interesting species.

L 50: are these taxonomic species? How many unique species sequences are identified in GenBank (e.g. 18S) ? This data should be included, as I suspect there are many species described in the past from morphology alone, with sequences missing—an area which may identify more (cryptic) species. So its worth mentioning an area for future improvement.

L 65: ‘rare taxa’ is misleading, as tardigrades are ubiquitous. Swap this wording to ‘understudied’ or ‘poorly investigated taxa’

L 76: again these are records without molecular data? This should be highlighted, since sequences can be compared to global diversity assessments- to show either endemism or (more likely) global dispersal.

L 85 what are ‘scientific outreach journals’ ? are these popular media publications?

L 99 odd wording, and "hierarchized”

L 101 add ‘potentially’ before restricted (see other comments about restriction from this dataset”

L 134-137- this is either brilliant or extremely poor methods. Justify why you have chosen say Rotifera (a similar sized microbial eukaryote which can form a resilient state enhancing potential dispersal and stress survival?). Extrapolating from one taxa to another is at best just a guess, and this must be mentioned.  Perhaps tardigrade diversity is 10x that due to tun formation. Or 10x less due to tun formation and global spread leading to far less speciation. Your dataset has nothing to say on this, but it should be discussed and present in conclusions.

L172: you can never calculate number of authors participating in taxonomic investigations. Not simply because many text can be missed (especially before ~1940), but because much investigations are made which are not successful, therefore never recorded- so, actual number of time spent is always going to be far higher than estimates here. Discuss this or omit any values.

L 217: figure 1 is excellent. It should be highlighted in text how many large regions of Mexico remain uninvestigated- which in turn should stimulate future research and microbial studies in the country. This should be a major conclusion of the paper.

L 222 poor wording.

L 232-244 This is showing that many species are recorded from world regions, and will likely be found as sampling efforts (in Mexico and beyond) intensify.

L 250 to 267. See comment above- this needs to be removed or completely reworked to align with the data presented (e.g. cannot prove actual vegetation type used by sampled tardigrades).

 L 282. “could be found in the country” is very different than could thrive in the country. Do your data examine only what can be detected, or what is actually present there? It is unclear if Mexico is unsuitable to any species of tardigrades (except for some adapted for polar or deep sea living).

L 309. Clench model. This needs to be presented in a methods section.

L 342. An interesting figure. It is difficult to say however that authors involved in tardigrades were actively conducting intensive sampling campaigns during any one time period. For example, an author could spend 4 years looking at tardigrades- with 1 year being field work, 1 year being lab work, and another being PhD write up in which no sampling occurred. Therefore, this would be a misleading data point. This must be addressed in the text.

L354-358: this is completely nonsense. You are extrapolating far beyond the dataset or logic. It is equally possible that it shall take 1,000 years to find the most cryptic tardigrade. More likely, there will be 1 scientist with a few students who record 50 species in 1 season. Your data cannot comment on this, so it must be removed.

L 367: “2,714 samples” is very misleading. This could be conducted in a series of several months in the wrong regions and yield nothing. So this number is arbitrary from a poor predictor. It needs to be removed or acknowledge the limitations.

L398 or somewhere similar you should include the molecular sequence issue such that many of these species lists (especially those not recording species) seem to lack such data- a major deficiency moving forward in understanding true diversity. E.g. discuss 18S rRNA gene or whatever biomarker is most popular within tardigrades.

L470: presumably these lack molecular sequences? Again, Genbank etc should be referenced to see how many unique species tree out here. (although errors of course exist in this data base also!).

L478- perhaps major diversity exists in Mexican waters (some of which is very deep). This could be discussed further, and point towards a future direction for the field.

L 542- 550. This section should be removed. The numbers appear arbitrary (even from the cited paper). This only acts to discourage others from the field. I have described species for far far less than the number presented here, and since the authors argue that some authors do the bulk of the work, an entire lab set up can be divided by each species number making the total investment far less (and of course, other duties are part of this calculation).

L 560- 561- this is should be removed. There are no evidence supporting species of tardigrades can go extinct, indeed they may outlast us all. Also, extinct in Mexico due to habitat loss is not the same as extinct from Earth. Rework this section to show that habitat loss will lessen the understanding of potential diversity in Mexico if you like.

The conclusion is far too short.

L576: ‘confirm their endemicity’ is unlikely due to all the points the authors have made about difficulty in detecting species

Author Response

We are resubmitting our original research article entitled “A strategy to provide a scenario of Mexican biodiversity of Tardigrada: species list, description of their distribution patterns, estimation of unknown species, and taxonomic effort to complete national inventory.” for its consideration in Diversity — Open Access Journal.

We are pleased to receive the revisions and appreciate the comments made, which together with improved the manuscript. Every revision was highlighted, we provide a detailed revision citing the line number and exact change.

Revisor 3

We have addressed the comments received, in general, we can say that the species current absences are taken in such a way that the current absences and the current distribution shown are a hypothesis, which may change when evidence becomes available. In the introduction, we add information on the tolerance of tardigrades to extreme environmental conditions, in the context of dispersion and distribution of the species. Regarding the vegetation section, it has been eliminated, the same as figure 2, based on the level of scale that you brought to our attention, which we agree on. We have added phase-contrast microscopy images of some of the species of following the categories: wide distribution, Nearctic distribution, only registered in America, and only register in type locality (in Mexico).

Finally, the conclusion has been expanded based on the information added in the text.

Below we show the changes made in each of the cases.

Major issues:

Restriction to one region is impossible to say due to undersampling, a problem well known in microbiology (e.g. see the debate within ciliates). The authors state how many new species were recorded or described from Mexico in the last ten years, and how before 2011 little was known—this is proof of undersamplilng, so imagine what else may be found within the next 10 years? (especially as molecular databases increase). Tardigrades thought to be restricted to 1 region may be found thriving in all after intensive sampling efforts are increased. Difficult issue due to the vast regions of Mexico and few taxonomists/ scientists working in these areas. This must be addressed in the manuscript as recording current absence is NOT proof of absence at a regional scale in the microbial world. Response: We agree, therefore we have added the following paragraph: Page 39, Line 113-117 The sites where there are no records of presence are not evidence of absence on a regional scale, but rather of little or no collection effort in these areas, so we cannot call them restricted or endemic species.

Missing from the introduction is standard details about how Tardigrades are extremely environmentally resilient (famous for surviving in 0g, radiation, desiccations, heat etc etc). This is necessary not only to give interest to the reader but is relevant as this paper is on to some extent species dispersal. The tardigrade survivability is directly related to its dispersal- and therefore it seems unlikely that a tardigrade could be endemic due to its ability to thrive in so many extremes. Discuss the resilient forms (tun state) along with the extremes they can tolerate compared with global regions. Response: We agree, therefore we have added the following paragraph: Page 1, Line 41-45. Given the wide variety of environments in which they can live, tardigrades have devoted significant amounts of energy to specific adaptive strategies for survival, such as resting stages, generally defined as dormant, which involve a temporary suspension of active life, reduced or suspended metabolism and arrested development (Guidetti et al 2011). The great resilience of these organisms has allowed them to disperse through air and water; and recently possible evidence that birds play a role as long-distance dispersers of tardigrades has been found (Robertson et al 2020).

Since you are estimating what species may be found in Mexico, you must discuss how they could get there, and issues surrounding actually detecting them. Response: We agree, a paragraph was added in the conclusion explaining the question (pg. 24, lines 617-619).

L 250: 3.2.2 “vegetation types”. This entire section is flawed in that there is no evidence presented as to the actual type of vegetation from the tardigrade collection. Since these are microbial animals, they inhabit a diverse array of niches. Even if a GPS location records a ‘forest’ area for the collection- it could have easily been a pond, a puddle on a road from a jeep, or on an actual lichen which is irrelevant to tree type (or a sign post on the road). In addition, this can be simplified to collecting a tardigrade in a scientists backyard potted plant at a house, rather than a farmland the house may be on and recorded as on a GIS map of vegetation. I like what the authors are trying to show here, but unfortunately their data set cannot support such claims or type of actual habitat utilized by the taridgrades at collection (unless explicit data was given in the published literature, which was not indicated and would require extensive further review). This should be removed entirely or fully acknowledged as to the limitations expressed here for sampling microbes in micro niches.

Figure 2 also looks great, and this could be a meaningful contribution to the paper. But as I have pointed out here, a vague recording of a species from one area cannot be used as evidence for the exact vegetation type- and this is irrelevant as many species are found in freshwaters, or in ubiquitous mosses which thrive irrespective to overlaying ground vegetation classification. This needs to be removed or justified in a way currently not present in the text from the dataset. Response: It has been eliminated, the same as figure 2, based on the level of scale that you brought to our attention, which we agree on.

Also missing from the text are imagery of tardigrades. I realize that the authors may not have done any microscopy as part of this study, but it seems a big miss to not include a 1 figure plate (e.g. 2 to 4 images) showing the diversity of tardigrades, with actual images on such micro animals from Mexico. It seems that obtaining 4 photos for publication would be very easy, even if colleagues must be contacted. Response: We have added phase-contrast microscopy images of some of the species of following the categories: wide distribution, Nearctic distribution, only registered in America, and only register in type locality (in Mexico).

The authors in several instances do over interoperate their data to extremes, without discussing it or its limitations. L354-358 is an example of this, such that the data cannot comment on how long it could take a given scientist(s) to uncover unknown diversity. This could be presented as a discussion if argued correctly with limitations, but the authors do not present it in such a way, which currently it must be reworded heavily or removed. Response: We agree, and we have added a paragraph: However, when analyzing some of the most recent species lists, elaborated for certain regions of the country, such as those of Dueñas-Cedillo et al. [28] and Moreno-Talamantes et al. [3] show that in a systematized sampling it is possible to find between nine and eleven species, of which only one is new to science, which is equivalent to approximately 10% of the diversity (9.09-11.11%); thus, of the 216 missing species, only 10% of them could be new to science, so describing them would take about 18 years. It should be noted that it is necessary to recognize the scope and limitations of this extrapolation, since in this case only the current rate of species description in Mexico has been evaluated and only the three existing working groups in the country are considered, so if the number of researchers interested in this group increases, this rate will increase over time. Another factor not considered is the taxonomic effort invested in research that has failed and has not allowed the description of new species, which would increase the time to discover those new species.

The conclusions is less than ~20 lines. This is far too short. Consider the points I have made above to add to the discussion, which you can then conclude with. Response: We agree, and add the following paragraphs: In tardigrades, the systematic has historically been based on morphological characters such as size and shape of the claws, organization of the buccopharyngeal apparatus, cuticular patterns, and egg morphology. However, these structures present intra-specific variation that has not been quantified with more precise methods; it is, therefore, necessary to use integrative methods using different sources of information, such as ecological data, molecular data from nuclear and mitochondrial DNA, and morphological characters. would be invaluable to future studies.

The best-sampled provinces in the country were TVBP, SMOrP, and SMOcP and the least explored were SMSP, PLP, and BBP, although in all provinces, records are concentrated in one or two states, leaving much territory to be explored, likewise remain unexplored two biogeographic provinces and 6 marine ecoregions, which in turn should stimulate future research and microbial studies in the country. Another factor to evaluate in terms of tardigrade diversity is the depth since in Mexico there are still deeper areas to be explored, such as the abyssal zones that are present in the Gulf of Mexico, the Caribbean, and the Atlantic Ocean.

After this revision, six taxa, expanded their known distribution and as sampling efforts intensify in Mexico and other regions of the world, this pattern of distribution spread will become increasingly common.

Minor edits:

  1. Line 19-20: this doesn’t make much sense. Add in “known” before tardigrade, otherwise, it sounds like they are invasive. Also, if taxa number is up, by definition so too is their diversity. Rethink this opener. Response: We agree, and we have made the corresponding change.

  1. L 21: add “for understanding” before Mexican. Response: We agree, and we have made the corresponding change.
  2.  
  3. L 23 add “and estimate” before ‘the taxonomic’. Response: We agree, and we have made the corresponding change.

  1. L 24 reword to not start a sentence off with a number. Response: We agree, and we have made the corresponding change.

  1. L 31 missing a period before ‘Diversity’. Response: We agree, and we have made the corresponding change.

  1. L32 add ‘of Tadrigrades’. Response: We agree, and we have made the corresponding change.

  1. L 37: why 17th? Was this their list, or is Mexico #17 out of say 20 or 50? Seems an off number value. Response: We agree, we have changed the wording of the text, in which we specifically refer to the fact that Mexico is in fourth place in terms of species richness, with 10 percent of the world's species living in its territory, which corresponds to little more than one percent of the earth's surface.

  1. L 45-50: Consider moving this tardigrade info bit to the beginning of the introduction, then introduce Mexico as a species rich area which has not yet been studied for these interesting species. Response: We agree, and we have made the corresponding change.

  1. L 50: are these taxonomic species? How many unique species sequences are identified in GenBank (e.g. 18S) ? This data should be included, as I suspect there are many species described in the past from morphology alone, with sequences missing—an area which may identify more (cryptic) species. So its worth mentioning an area for future improvement. Response: We carry out the corresponding search and add the pertinent information

  1. L 65: ‘rare taxa’ is misleading, as tardigrades are ubiquitous. Swap this wording to ‘understudied’ or ‘poorly investigated taxa’. Response: We mean “rare” in the ecological sense, so we change “rare” to taxon with low abundance.

  1. L 76: again these are records without molecular data? This should be highlighted, since sequences can be compared to global diversity assessments- to show either endemism or (more likely) global dispersal. Response: As we mentioned in the text, there are few taxonomists dedicated to the study of this group, and the documented species lack molecular data, which, together with descriptions and analysis of their distribution patterns, could allow comparisons of global diversity, to show endemism or dispersal.

  1. L 85 what are ‘scientific outreach journals’ ? are these popular media publications?. Response: We refer to the revista de divulgación científica.

  1. L 99 odd wording, and "hierarchized”. Response: We agree, we refer to the organization of biogeographical regions. We have changed the wording.

  1. L 101 add ‘potentially’ before restricted (see other comments about restriction from this dataset”. Response: We agree, and we have made the corresponding change.

  1. L 134-137- this is either brilliant or extremely poor methods. Justify why you have chosen say Rotifera (a similar sized microbial eukaryote which can form a resilient state enhancing potential dispersal and stress survival?). Extrapolating from one taxa to another is at best just a guess, and this must be mentioned. Perhaps tardigrade diversity is 10x that due to tun formation. Or 10x less due to tun formation and global spread leading to far less speciation. Your dataset has nothing to say on this, but it should be discussed and present in conclusions. Response: The Rotifera case mentioned in the text is just an example of how the calculation shown in Table 2 was performed. We agree with your comment and mention the limitations of our methods, which are mentioned in the discussion and conclusions.

  1. L172: you can never calculate number of authors participating in taxonomic investigations. Not simply because many text can be missed (especially before ~1940), but because much investigations are made which are not successful, therefore never recorded- so, actual number of time spent is always going to be far higher than estimates here. Discuss this or omit any values. Response: As we mentioned in the materials and methods, we rely on published work and research. However, his observation was heeded, and the necessary information was added to the discussion.

  1. L 217: figure 1 is excellent. It should be highlighted in text how many large regions of Mexico remain uninvestigated- which in turn should stimulate future research and microbial studies in the country. This should be a major conclusion of the paper. Response: We agree, and we have made the corresponding change.

  1. L 222 poor wording. Response: We agree, and we have made the corresponding change.

  1. L 232-244 This is showing that many species are recorded from world regions, and will likely be found as sampling efforts (in Mexico and beyond) intensify. Response: We agree, we emphasize the suggested idea.

  1. L 250 to 267. See comment above- this needs to be removed or completely reworked to align with the data presented (e.g. cannot prove actual vegetation type used by sampled tardigrades). Response: We agree, we have deleted the section corresponding to vegetation.

  1. L 282. “could be found in the country” is very different than could thrive in the country. Do your data examine only what can be detected, or what is actually present there? It is unclear if Mexico is unsuitable to any species of tardigrades (except for some adapted for polar or deep sea living). Response: We agree, we have changed the wording.

  1. L 309. Clench model. This needs to be presented in a methods section. Response: In the methods section we mention the assumptions of the Clench model, as well as the cases in which it can be used.

  1. L 342. An interesting figure. It is difficult to say however that authors involved in tardigrades were actively conducting intensive sampling campaigns during any one time period. For example, an author could spend 4 years looking at tardigrades- with 1 year being field work, 1 year being lab work, and another being PhD write up in which no sampling occurred. Therefore, this would be a misleading data point. This must be addressed in the text. Response: The objective of this figure is to show the changes in the number of described species and the number of authors involved in those descriptions over time, and thus temporarily contextualize the descriptions made in Mexico.

  1. L354-358: this is completely nonsense. You are extrapolating far beyond the dataset or logic. It is equally possible that it shall take 1,000 years to find the most cryptic tardigrade. More likely, there will be 1 scientist with a few students who record 50 species in 1 season. Your data cannot comment on this, so it must be removed. Response: We agree, the paragraph was considered part of the discussion, contextualizing the number of years for the description of potential species concerning the current trend. Likewise, we mention the limitations of our estimation.

  1. L 367: “2,714 samples” is very misleading. This could be conducted in a series of several months in the wrong regions and yield nothing. So this number is arbitrary from a poor predictor. It needs to be removed or acknowledge the limitations. Response: This number is the result of the Clench model, which already includes failed samples.

  1. L398 or somewhere similar you should include the molecular sequence issue such that many of these species lists (especially those not recording species) seem to lack such data- a major deficiency moving forward in understanding true diversity. E.g. discuss 18S rRNA gene or whatever biomarker is most popular within tardigrades. Response: We agree, this information is discussed in the corresponding section.

  1. L470: presumably these lack molecular sequences? Again, Genbank etc should be referenced to see how many unique species tree out here. (although errors of course exist in this data base also!). Response: We agree, this information is discussed in the corresponding section.

  1. L478- perhaps major diversity exists in Mexican waters (some of which is very deep). This could be discussed further, and point towards a future direction for the field. Response: We agree, this information is discussed in the corresponding section.

  1. L 542- 550. This section should be removed. The numbers appear arbitrary (even from the cited paper). This only acts to discourage others from the field. I have described species for far far less than the number presented here, and since the authors argue that some authors do the bulk of the work, an entire lab set up can be divided by each species number making the total investment far less (and of course, other duties are part of this calculation). Response: We have removed the section.

  1. L 560- 561- this is should be removed. There is no evidence supporting species of tardigrades can go extinct, indeed they may outlast us all. Also, extinct in Mexico due to habitat loss is not the same as extinct from Earth. Rework this section to show that habitat loss will lessen the understanding of potential diversity in Mexico if you like. Response: We agree, we have reworked this section.

  1. The conclusion is far too short. Response: We agree, we have reworked this section

  1. L576: ‘confirm their endemicity’ is unlikely due to all the points the authors have made about difficulty in detecting species. Response: We agree, we have changed the wording of this sentence throughout the text.

Reviewer 4 Report

Please find my comments attached in the file.

Author Response

We are resubmitting our original research article entitled “A strategy to provide a scenario of Mexican biodiversity of Tardigrada: species list, description of their distribution patterns, estimation of unknown species, and taxonomic effort to complete national inventory.” for its consideration in Diversity — Open Access Journal.

We are pleased to receive the revisions and appreciate the comments made, which together with improved the manuscript. Every revision was highlighted, we provide a detailed revision citing the line number and exact change.

Revisor 4

  1. line 24: "39 genera" instead of "43 genera" is more correct now - see Table 1 and Figs 1 and 2. But you have to take into account that the 40th edition of the Checklist was published already in February 2022 when you accessed Checklist so you have to apply 40th Checklist edition. For this reason you have to change the taxonomic status of several species (especially in Table 1 and some Figs): Adropion carolae is Guidettion carolae, Echiniscus kofordi is Kristenseniscus kofordi, Isohypsibius sattleri is Dianea sattleri, Macrobiotus furcatus is Minibiotus furcatus, Pseudechiniscus gullii is a synonym of Pseudechiniscus quadrilobatus So correct number of genera is 42 in your study. Response: We reviewed the 40th edition of the current checklist and recounted the genres, giving a total of 42.

  1. Introduction line 44: "tardigrades" instead of "Tardigrades" is correct.

Response: The paragraph was completely restructured; the word Tardigrades is no longer there.

  1. line 45: "1200 μm" instead of "1000 μm" is correct - see e.g. Nelson, D.R., Guidetti, R., Rebecchi, L., 2015: Phylum Tardigrada. In: Thorp, J., Rogers, D.C. (Eds.), Ecology and General Biology: Thorp and Covich's Freshwater Invertebrates, Academic Press, 347–380. Response: We reviewed the document and made the correction.

  1. lines 49-50: "at least 150 genera and more than 1350 species" instead of "at least 90 genera and more than 1300 species" is correct (153 genera and 1380 species are in the 40th Checklist edition you used according to date of accession in the list of references!). Response: We reviewed the listing and made the corresponding change.

Materials and Methods

  1. line 124: "onychophorans" instead of "onycophores" is correct. Response: We make the corresponding change.

  1. lines 130, 134, 279: "1380" instead of "1338" is current diversity according to Actual checklist (40th edition) which you should use!. Response: We reviewed the listing and made the corresponding change.

  1. line 135: "84 species" instead of "80 species" is correct (if we consider current diversity 1380 species according to 40th edition of the Checklist). Response: We recalculated to update the current diversity of tardigrades according to the 40th edition of the Checklist.

  1. line 136: "162 species" instead of "163 species" is correct. Response: We make the corresponding change.

  1. lines 172-174: Authors wrote "All species, year of description and corresponding authorities were listed from the 38th edition of actual checklist of tardigrade species [18] and new records until May 2021 [77]". According to list of references authors studied Actual checklist in 03.02.2022. So authors have to apply 40th edition of the Checklist which was published before that date! Response: We revised again and only based on the current 40th edition listing.

Results

  1. line 185: Halechiniscidae is not a subfamily. Please correct this throughout the text where you mentioned two subfamilies instead of one family and one subfamily. Response: We make the corresponding change.

  1. line 186: "42 genera" instead of "43 genera" - see my comment to the line 24. Response: We make the corresponding change.

  1. lines 187-195: Please add the information how many times four missing taxa (Eutardigrada, Diaforobiotus, Halechiniscus and Viridiscus) were recorded! In addition, please add such information for newly added genera Guidettion, Kristenseniscus and Dianea - see my comment to the line 24. And of course some data in this lines have to be recalculated as some genera will be transferred into others. Response: We reviewed the records and made the pertinent corrections.

  1. line 190: "Milnesioides" instead of "Milnesoides" is correct. Response: We make the corresponding change.

  1. line 192: I do not understand how it is possible that Isohypsibius was recorded only two times while according to Table 1 two species of the genus were recorded and at least two times this taxon was recorded only on generic level. Please correct what necessary. Response: We correct the corresponding data.

  1. line 194: I do not understand how it is possible Calcarobiotus was recorded only one time while according to Table 1 two species of the genus were recorded. Response: In the table there is only one species of Calcarobiotus.

  1. line 194: "Dactylobiotus" instead of "Dactylotobiotus" is correct. Response: We corrected the error and confirmed that in other sites it was correctly written.

  1. lines 220-221: Authors wrote "The list of Mexican tardigrades recorded and the integration of their national and global information of the distribution is presented in Table 1 [85–91]. " What is the source of data concerning global distribution of each taxon (column "Records and distribution") - as I know cited sources do not contain such data. Please add relevant citation. Response: We add the corresponding citation for each species.

  1. line 222: "58 taxa" instead of "68 taxa" is correct. Response: We make the corresponding change.

  1. line 243: Please correct "Caribe Sea". Response: We made the change in all the places where it was found to be misspelled.

  1. line 274: 15 taxa are in Table 2 (not 14 taxa). Response: We made the change on this line and other sites where that figure is mentioned.

  1. line 295: "onychophores" instead of "onycophores" is correct. Response: We make the corresponding change.

  1. line 252: "two" instead of "two 2" is correct. Response: The section of vegetation was completely removed.

  1. lines 306-307: Authors wrote "The slopes at the end of both curves are greater than 0.1, which indicates that the inventories are incomplete and unreliable (m genera = 0.089; m species = 0.78)". First it is necessary to say that 0,089 is not greater than 0.1. If I good understand this sentence authors meant linear ends of both curves. I modeled the ends of both curves in Excel and the results are: Response: We correct the statement: “The slope at the end of the curve at generic level is lower than 0.1, which indicates that the inventory can be considered sufficiently reliable (m genera = 0.089), although it is still incomplete; while the slope at the end of the curve at specific lever is greater than 0.1 which indicates that the inventory is incomplete and unreliable (m species = 0.78).

Discussion

  1. line 361: "42 genera" instead of "43 genera" - see my comment to the line 24. Response: We make the corresponding change.

  1. line 483: Please re-calculate the value 6.6% as I explained in note to the table 2. Response: We performed the corresponding calculation and obtained a value of 7.27.

  1. lines 520-521: Authors wrote "Unfortunately, in other countries within the region, as Panama, Venezuela, Bolivia, among many others, tardigrade species remain undiscovered." In Venezuela and Bolivia there were tardigrades discovered already [Kaczmarek, L, Michalczyk, L., McInnes, S.J., 2015: Annotated zoogeography of non-marine Tardigrada. Part II: South America. Zootaxa 3923 (1): 1-107.]! Response: We reviewed the literature and added instead Trinidad and Tobago, Belize, Guatemala, El Salvador and Honduras.

  1. line 550: How did you obtained 13,237,000 USD? I think 213 species multiplied with 61,000 USD per species is 12,780,000 USD. Response: This section was completely removed.

Conclusions

  1. line 566: "42 genera" instead of "43 genera" - see my comment to the line 24. Response: We make the corresponding change.

References

  1. line 637: Please add names of authors in the citation of the 40th Checklist edition. Response: We make the corresponding change.

  1. line 640: Please divide the word "Noveaugenre". Response: We make the corresponding change.

Tables and Figures

  1. Table 1: Please correct English/Latin in "Irlanda", "Italia", "Islandia", "Kerguelen Is. (India)"- not belonging to India, "Hymalayas", "caribe", "Dactylotobiotus" and "Broadely distributed". Response: We make the corresponding change.

  1. Table 1: Please add parentheses in taxa names when necessary and correct invalid names (g. Isohypsibius sattleri is in the genus Dianea, Astatumen trinacriae has different author and year,...) according to the 40th edition of the Checklist. Response: We make the corresponding change and review the typical land and authors of each species.

  1. Table 1: Authors centered the contents of each cell both vertically and horizontally, and this caused reading difficulties. For example, assigning the contents of the "Records and distribution" column to specific species is problematic. Therefore, I strongly suggest to align the contents of each column horizontally to the left (the "Reference" column may remain centered) and vertically to the top. Response: We reorganized the information in the table to make it easier to read.

  1. Table 1: Please check Terra typica for each species again (g. Pseudechiniscus juanitae was described from Brazil and not from Costa Rica). Response: We corrected the data and made sure that all species had the correct information.

  1. Table 1: What is the source of data in the column "Records and distribution"? Please cite relevant source(s). Response: We corrected; each species has the corresponding citations.

  1. Table 2: Please align first column to the left and others to the right. Please unify number of decimal places in each column. Response: We make the corresponding change.

  1. Table 2: 013825 arthropods species is apparently wrong number. Please correct it. Response: We make the corresponding change, the correct number is 1013825.

  1. Table 2: I consider average percentage 6.61% calculated from all percentages as wrong way of calculation. Better is sum of species known in Mexico from 15 taxa divided by sum of species known in the world from the same 15 taxa. Please re-calculate the percentage and subsequently "Actual" and "Estimation" (or Estimated?) richness in the Table 2. Response: We recalculated the average percentage of 7.72%, we also changed Estimation to Estimated.

  1. Table 2: Actual richness according to the Checklist (40th edition) is 1380 and not 1338. Please respect this number and re-calculate number of tardigrades in Table 2! Response: We make the corresponding change and recalculate the number of tardigrades.

  1. Table 3: Known global richness of tardigrades is 1380 species and not 1338 ones. Response: We make the corresponding change.

  1. Table 3: Please explain why did you excluded Protura and Diplura from the calculation of "Range of species estimated" (as you wrote in lines 294-295). Response: We add the explanation of why certain groups were removed from the range.

  1. Table 3: Range 74-505 is apparently wrong (you probably meant 74-254). On the other hand, you mentioned this range was calculated also from copepods (line 295) the range apparently should be 7-254. In any case you have to re-calculate it because of known richness 1380 species (and not 1338 ones). Response: We recalculate the data and correct the table.

  1. Table 3: Probably range 113-225 is also wrong. Apparently range 14-505 is correct (when you used data also from copepods as you wrote in line 295). Response: We excluding the values of the taxa that quantified less than 76 species, that is the number of actual tardigrades records in Mexico obtained in this work.

  1. Table 3: For me it is not clear how you obtained values 95 and 189. Why not 87 and 176? In any case mean value 87 has to be re-calculated based on 1380 known species. Response: We recalculate the data and correct the table.

  1. Table 4: For me it is not clear how authors calculated number of missing samples (named as Samples needed in Table 3) and Recorded proportion (in Table 3 named as effectively record). In any case value 572 in this Table and the value 527 (in Table 3) are not same -please correct what necessary. Response: Missing samples are obtained from the Clench model equation, we add the description of this in the methods: Subsequently, the quality of the sampling n=a/(1+b·n)2 and the sampling effort required to record 95% of the fauna n95 =0.95/ [b(1-0.95)] was estimated with the Clench´s equation Sn=a·n/(1+b·n), this model assumes that the probability of adding new species to the inventory increases as more time is spent in the field and is recommended for large area studies and for protocols [81].

  1. 2: This figure is haerdly readable - to find any type of vegetation according its colour is very difficult. I strongly suggest replace this picture with additional column for Vegetation types into the Table 1. Response: The vegetation section was completely removed.

  1. Figure 3: The level of "44 genus" should be named as "44 genera". Response: We make the corresponding change.

  1. Figure 3: The whole denominator in both equations must be in parentheses (not just its final product)!! Response: We make the corresponding change.

  1. Figure 3: Only the modeled curves (fitted lines) are in the picture, but the measured values are missing! Missing points must be added to the graphs. Response: We make the corresponding change.

Round 2

Reviewer 3 Report

Tardigrade review #2

The authors have provided extensive revisions from their original draft on Tardirade diversity in Mexico. They provide a literature review of tardigrade describes, as well as a map of described species within Mexico. The paper still falls short due to several over extrapolations in the data. I am unsure if the authors issues here are difficulties in language, or true methodological error? In any case, I provide several major issues, as well as a couple of minor issues. Overall this paper can be a good contribution to species list and diversity studies, but needs rewording etc. 

I will spend no time in this review researching the taxonomic classifications or looking up species numbers of names and phylogeny (e.g. ALL of table 1, and values within table 2)- I will leave this to the special issue editor and other reviewers, and thus defer to their ultimate decision. I do note that many changes in this area are present in this version.

Major issues:

L 160-168 I still do not find an acceptable explanation of an unknown diversity value. Much of the 1380 species are likely from old records and morphology alone, and thus synonyms exist. To a similarly divergent answer- many species remain cryptic and unidentified at a global level. With these two facts in mind, an estimate of diversity (the authors give 6.1%) is virtually no better than a random probability guess, as the true value could be .5 or 20% with as much evidence given.

L200-202 I remain unconvinced that an actual value of taxonomists and their time required can ever be quantified for a region the size of Mexico. The discrepancies between an expert dedicated to the task vs an intern is apparent, and shall not allow for a real value to be given. Phrase this section more as an estimate or a guess rather than a fact established from the data here.

L648-656- I completely disagree with this new addition. The made up rate of 1.66 species per year with 215 alleged missing species cannot give an accurate estimate. It could be 100 years, or it could be 1 really good PhD student with molecular funding to identify 50 species in a 6 month season. The data presented here gives no insights into this problem, and should be worded accordingly.

The addition of the tardigrade micrographs enhances the text, well done.

Minor issues:

Please ensure that Figure 1 is a high resolution TIFF file for any subsequent publication. The PDF provided for review is far too blurry for use. Bump up the res to at least 10MB.

L137 switch arrange to range.

L 182 should be ‘theses’

Figure 2 L237 needs explanation: E.g. ‘micrographs of tardigrades taken in vivo from Mexico…”

The PDF from the journal has made the gray scale background different odd shaded. Ensure the original file (TIFF) is of high quality for any usage.

L365-372 this is not clear, and the slope predictions are not proof of their accuracy.

L606-607 these numbers are only a guess, and should be identified as such.

L650 “a pair of decades’ is awkward wording.

There are some English language issues (especially in the new text) which need to be addressed. I will leave this to the capable editors to correct.

Author Response

Comments and Suggestions for Authors

Tardigrade review #2

The line numbers marked by the reviewer do not coincide with the numbering of the version that we uploaded, so it was difficult to relate the number of lines with the comments; in the comment on line L606-607 we could not find the paragraph to which it referred. The remained comments, were solved.

We made the requested adjustments mainly related to the richness estimation methodology and taxonomic effort, as well as the proposed minor changes.

Major issues:

L 160-168 I still do not find an acceptable explanation of an unknown diversity value. Much of the 1380 species are likely from old records and morphology alone, and thus synonyms exist. To a similarly divergent answer- many species remain cryptic and unidentified at a global level. With these two facts in mind, an estimate of diversity (the authors give 6.1%) is virtually no better than a random probability guess, as the true value could be .5 or 20% with as much evidence given. Response: We agree with the reviewer's comment. We assume that species are dynamic entities that change over time and that the description of a species constitutes a working hypothesis in light of the evidence found at that time; based on this it is not possible to have a precise value of diversity. Therefore we have added the following paragraph: Since most of the 1380 species of tardigrades known to date have been described from morphological data, and current integrative approaches argue that these attributes are not sufficient to support and recognize them, it is likely that there are synonymous or cryptic species in this number. For this reason, calculations obtained from methods that consider the current number of species should be taken as a conservative figure, which generally only estimates the possible diversity of Mexican tardigrades.

L200-202 I remain unconvinced that an actual value of taxonomists and their time required can ever be quantified for a region the size of Mexico. The discrepancies between an expert dedicated to the task vs an intern is apparent, and shall not allow for a real value to be given. Phrase this section more as an estimate or a guess rather than a fact established from the data here. Response: We agree with your observation, so we add this paragraph to the methods section: We do not attempt to assign a deterministic value to the number of authors and the time required to inventory the Mexican tardigrade fauna, since these variables depend on numerous factors that were not considered in the present study (e.g., the level of experience of taxonomists, the number of research groups and their infrastructure (national or international), the economic resources invested, the rate of habitat loss, are factors that could considerably affect the rate of species description). Rather, it should be considered that our estimates of potential diversity are based on biased samples from a few regions within the country and that most of the Mexican provinces have few or no records, particularly provinces characterized by their high species richness, which in turn could contribute significantly to the discovery of new taxa of Tardigrades in the country.

L648-656- I completely disagree with this new addition. The made up rate of 1.66 species per year with 215 alleged missing species cannot give an accurate estimate. It could be 100 years, or it could be 1 really good PhD student with molecular funding to identify 50 species in a 6 month season. The data presented here gives no insights into this problem, and should be worded accordingly. Response: We understand that this information may be speculative, so the section was removed.

The addition of the tardigrade micrographs enhances the text, well done.

Minor issues:

Please ensure that Figure 1 is a high resolution TIFF file for any subsequent publication. The PDF provided for review is far too blurry for use. Bump up the res to at least 10MB. Response: The resolution of the images is calibrated at 330dpi as requested by the magazine, hopefully they will look optimal in the final version.

L137 switch arrange to range. Response: We make the change in the word

L 182 should be ‘theses’. Response: We make the change in the word

Figure 2 L237 needs explanation: E.g. ‘ of tardigrades taken in vivo from Mexico…”micrographs.  Response: We add the following: Figure 2. PCM micrographs of some species of tardigrades present in Mexico.

The PDF from the journal has made the gray scale background different odd shaded. Ensure the original file (TIFF) is of high quality for any usage. Response: The resolution of the images is calibrated at 330dpi as requested by the magazine, hopefully they will look optimal in the final version.

L365-372 this is not clear, and the slope predictions are not proof of their accuracy. Response: We assume that this comment refers to the part of the results where the slope of the curve is mentioned, however, in these lines the quality and reliability of the inventory is mentioned but not its accuracy, however, in the discussion section we added a paragraph responding to this: The discovery curve method determined that for both genera and species the inventories are incomplete, because 22% of the genera and 61% of the species have yet to be discovered; however, estimates of the number of species derived in this way often lack associated margins of error, making it impossible to objectively assess their accuracy.

L606-607 these numbers are only a guess, and should be identified as such. Response: We were unable to relate the lines to the commentary.

L650 “a pair of decades’ is awkward wording. Response: We delete the section (see above).

There are some English language issues (especially in the new text) which need to be addressed. I will leave this to the capable editors to correct

Reviewer 4 Report

The authors have greatly improved their manuscript, but there are still fundamental errors (other than minor ones such as "specific rever" on line 370) that need to be corrected. I mean not respecting the ICZN as the authors ignore the use of parentheses in the names of species (compare, for example, the names in Table 1 with the names of the same species in the Actual checklist of Tardigrada species, e.g. Pilatobius nodulosus (Ramazzotti, 1957) i.e. author and year in paretheses!). It is also not allowed to write the name of the family in italics (e.g. Halechiniscidae in line 231) or the species name in capital letters (e.g. Calohypsibius cf. Ornatus in Table 1). The authors respected my other comments, with insignificant exceptions. Therefore, I do not recommend publishing the manuscript until the inaccuracies in the manuscript have been corrected.

Author Response

Comments and Suggestions for Authors

Tardigrade review #4

We revised the Actual checklist of tardigrades to ensure that the nomenclature of the species and authors was correct.

The authors have greatly improved their manuscript, but there are still fundamental errors (other than minor ones such as "specific rever" on line 370) that need to be corrected. I mean not respecting the ICZN as the authors ignore the use of parentheses in the names of species (compare, for example, the names in Table 1 with the names of the same species in the Actual checklist of Tardigrada species, e.g. Pilatobius nodulosus (Ramazzotti, 1957) i.e. author and year in paretheses!). It is also not allowed to write the name of the family in italics (e.g. Halechiniscidae in line 231) or the species name in capital letters (e.g. Calohypsibius cf. Ornatus in Table 1). The authors respected my other comments, with insignificant exceptions. Therefore, I do not recommend publishing the manuscript until the inaccuracies in the manuscript have been corrected. Response: We corrected the nomenclature in table one so that the species names and their authorities are written in the same way as in the Actual checklist of Tardigrada species, with the names of the species authors in parentheses in the species; Dipodarctus cf. subterraneus (Renaud-Debyser, 1959), Cornechiniscus lobatus (Ramazzotti, 1943), Kristenseniscus kofordi (Schuster & Grigarick, 1966), Pseudechiniscus suillus (Ehrenberg, 1853), Viridiscus viridis (Murray, 1910), Viridiscus viridissimus (Péterfi, 1956), Calohypsibius cf. ornatus (Richters, 1900), Doryphoribius evelinae (Marcus, 1928), Doryphoribius flavus (Iharos, 1966), Dianea sattleri (Richters, 1902), Adropion scoticum (Murray, 1905), Diphascon pingue (Marcus, 1936), Guidettion carolae (Binda & Pilato, 1969), Hypsibius cf. convergens (Urbanowicz, 1925), Isohypsibius sculptus (Ramazzotti, 1962), Pilatobius nodulosus (Ramazzotti, 1957), Calcarobiotus cf. polygonatus (Binda & Guglielmino, 1991), Macrobiotus cf. acadianus (Meyer & Domingue, 2011), Macrobiotus terminalis (Bertolani & Rebecchi, 1993), Mesobiotus coronatus (de Barros, 1942), Mesobiotus diffusus (Binda & Pilato, 1987), Mesobiotus harmsworthii (Murray, 1907), Minibiotus cf. intermedius (Plate, 1888), Minibiotus furcatus (Ehrenberg, 1859), Paramacrobiotus areolatus (Murray, 1907), Paramacrobiotus richtersi (Murray, 1911), Ramazzottius baumanni (Ramazzotti, 1962), Ramazzottius cf. oberhaeuseri (Doyère, 1840), and Diaforobiotus islandicus (Richters, 1904).

We also solved the problem with the capital letters after cf. as in: Pseudechiniscus cf. juanitae de Barros, 1939, Calohypsibius cf. ornatus (Richters, 1900). In addition, the names of the authors were corrected in: Minibiotus citlalium Dueñas-Cedillo & García-Román, 2020 in Dueñas-Cedillo et al. 2020, and a spacing error in Doryphoribius dawkinsi Michalczyk & Kaczmarek, 2010.

Finally, the error in the family Halechiniscidae, which was written in italics, was corrected.